



# 1 The spectral signature of cloud spatial structure in shortwave

# 2 irradiance

**Shi Song[1,2], K. Sebastian Schmidt[1,2], Peter Pilewskie[1,2], Michael D. King[2], Andrew K.**
**Heidinger[3], Andi Walther[3], Hironobu Iwabuchi[4], Odele M. Coddington[2]**
[1] Department of Atmospheric and Oceanic Sciences, University of Colorado, Boulder, CO,
USA
[2] Laboratory for Atmospheric and Space Physics, University of Colorado, Boulder, CO, USA
[3] NOAA Center for Satellite Applications and Research, Madison, WI, USA
[4] Center for Atmospheric and Oceanic Studies, Tohoku University, Japan
**Abstract**
We found that cloud spatial structure manifests itself as spectral signature in shortwave irradiance
fields – specifically in transmittance and net horizontal photon transport in the visible and near-
ultraviolet wavelength range. In this paper, we demonstrate this through radiative transfer
calculations with cloud imagery from a field experiment, and show that such three-dimensional
effects may occur on scales up to 60 kilometers. Neglecting net horizontal photon transport leads
to a transmittance bias on the order of ±12-19% even at the relatively coarse spatial resolution
of 20 kilometers, and of more than ±50% for 1 kilometer. This poses a problem for radiative
energy budget estimates from space because the bias for any pixel depends on its spatial context
in a non-trivial way. The key for solving this problem may lie in the spectral dimension, since we
found a robust correlation between the magnitude of net horizontal photon transport ($H$) and its
spectral dependence (slope). It is scale-invariant and holds for the entire pixel population of a
domain. This was at first surprising given the large degree of spatial inhomogeneity, but seems to
be valid for any cloud field. We prove that the underlying physical mechanism for this
phenomenon is molecular scattering in conjunction with cloud inhomogeneity. On this basis, we
developed a simple parameterization through a single parameter $\varepsilon$, which quantifies the
characteristic spectral signature of spatial heterogeneities. In a companion paper, we will show



that it is accompanied by spectral radiance perturbations, which can be detected from multi-
spectral imagers and may be translated into bias reductions for cloud radiative effect estimates in
the future.

## 1. Introduction

Determining cloud radiative effects for scenes with a high degree of spatial complexity
remains one of the most persistent problems in atmospheric radiation, especially at the surface
where satellite observations can only be used indirectly to infer energy budget terms. In the
shortwave (solar) spectral range, it is especially challenging to derive consistent albedo,
absorption, and transmittance from spaceborne, aircraft, and ground-based observations for
inhomogeneous cloud conditions (Kato et al., 2013; Ham et al., 2014). This problem is closely
related to the long-debated discrepancy between observed and modeled cloud absorption
(Stephens et al., 1990) since energy conservation for a three-dimensional (3D) atmosphere
(Marshak and Davis, 2005, Eq. 12.13)
$R + T = 1 - (A + H)$            (1)
connects reflectance[1] $R$, transmittance $T$, and absorptance $A$ of a layer. The term $H$ accounts for
lateral net radiative flux from pixel to pixel (which we will call net horizontal photon transport[2]).
Out of necessity, most algorithms for deriving $R$, $T$, and $A$ from passive imagery inherently
presume isolated pixels by relying on one-dimensional (1D) radiative transfer (independent pixel
approximation) which does not reproduce $H$. Net horizontal photon transport has therefore long
been a common explanation not only for inconsistencies between measured and calculated
broadband cloud absorption (Fritz and MacDonald, 1951; Ackerman and Cox, 1981) but also for
remote sensing artifacts (Platnick, 2001).
Observational evidence for this explanation emerged with the availability of spectrally

---

[1] albedo for reflected irradiance (flux density).

[2] Our use of the term "photon" is rooted in Monte Carlo radiative transfer.





resolved aircraft measurements of shortwave irradiance (Solar Spectral Flux Radiometer, SSFR:
Pilewski et al., 2003). Schmidt et al. (2010) derived *apparent absorption*, the sum of $A$ and $H$,
from irradiance measurements aboard the NASA ER-2 and DC-8 aircraft that flew along a
collocated path above and below a heterogeneous anvil cloud during the Tropical Composition,
Cloud and Climate Coupling Experiment (TC$^4$) (Toon et al., 2010).  The spectral dependence of
apparent absorption as well as its pixel-to-pixel variability showed that in absolute terms, $H$ at
visible wavelengths (where cloud and gas absorption are negligible) can assume similar values as
the absorbed irradiance $A$ at near-infrared wavelengths (where $|H| \ll A$). Horizontal photon
transport thus has the potential to mimic substantially enhanced absorption in broadband
measurements. Three-dimensional (3D) calculations confirmed the measurements, and radiative
closure was achieved within measurement and model uncertainties without invoking proposed
enhanced gas absorption (Arking, 1999) or big cloud droplets (Wiscombe et al., 1984). The
results also suggested that the overestimation of absorption would persist even when averaging
over long distances as proposed by Titov (1998). This is simply because radiation flight legs are
often preferentially targeted at cloudy regions ($\langle H \rangle > 0$) and do not adequately sample clear-sky
areas where photons are depleted ($\langle H \rangle < 0$), which is interpreted as *apparent emission* in
measurements.
Perhaps the most significant finding by Schmidt et al. (2010) was the distinct spectral
shape of $H$ from the near-ultraviolet well into the visible wavelength range, leading to the notion
of "colored" net horizontal photon transport (Schmidt et al., 2014).[3] Strategies for mitigating the
overestimation of cloud absorption (Ackerman and Cox, 1981; Marshak et al., 1999) require that
$H$ be more or less constant in the visible wavelength range (Welch et al., 1980), and so the

---

[3] A previous study addressing horizontal photon transport from an energy budget point of view (Kassianov and Kogan, 2002) had focused on the wavelength range of 0.7-2.7 μm, specifically to avoid molecular scattering at shorter wavelengths.



discovery of the spectral dependence of $H$ suggested that they should be applied with caution.[4]

2        Further analysis of the relationship between cloud structure and its spectral signature,

presented here, revealed a surprisingly robust correlation between the *magnitude* of $H$ and its
*spectral slope, dH/dλ*. In the course of this paper, we provide evidence for molecular scattering as
the physical mechanism behind this correlation and develop a simple parameterization based on
this knowledge. In an accompanying paper (Song et al., 2015), we will demonstrate that cloud
spatial inhomogeneities also manifest themselves in spectral *radiance* perturbations via *dH/dλ*,
which can be used for deriving $H$ correction terms for cloud radiative effects of inhomogeneous
scenes from space-borne observations.

10        We complete our paper by examining at which spatial aggregation $H$ can be ignored and

whether the discovered correlation between $H$ and *dH/dλ* is scale invariant. Finally, we consider
the ramifications of our findings on the shortwave surface energy budget and find that while
cloud transmittance biases may be significant even after spatial averaging, they are also
accompanied by spectral perturbations similar to the ones we encountered for $H$. These biases
may thus be detectable and correctable using adequate ground-based radiometers.

16        Following this introduction, we provide definitions of relevant terms and explain how $H$

relates to top-of-atmosphere (TOA) and surface cloud radiative effects (CRE). We then discuss
the data and model calculations that lay the basis for our study (Sections 3 and 4). In section 5,
we discuss the correlations between $H$ and *dH/dλ*, followed by the underlying physical
mechanism and parameterization presented in Section 6. The discovered relationship is then
examined as a function of spatial scale (Section 7) and interpreted in terms of the surface CRE
(Section 8). In the conclusions, we discuss the significance of our findings and propose multi-
spectral or spectral techniques for deriving first-order correction factors in CRE estimates from
space, aircraft, and from the surface that may render 3D calculations unnecessary.

---

[4] For example, Marshak et al. (1999) in their conditional sampling technique require that $H = 0$
for at least two different wavelengths. Kindel et al. (2011) applied such a modified scheme for
boundary layer clouds.





**2. Net horizontal photon transport and cloud radiative effect**

2       The instantaneous radiative effect of any atmospheric constituent is the difference of net

irradiance (flux density) in its presence (all-sky) and absence (clear-sky). For clouds, we define

$$CRE_\lambda = \left[ \frac{\left( F_\lambda^\downarrow - F_\lambda^\uparrow \right)_{all-sky}}{F_\lambda^{\downarrow,TOA}} - \frac{\left( F_\lambda^\downarrow - F_\lambda^\uparrow \right)_{clear-sky}}{F_\lambda^{\downarrow,TOA}} \right] \times 100\%, \tag{2}$$

where $F_\lambda^\downarrow$ and $F_\lambda^\uparrow$ are downwelling and upwelling irradiance and their difference is net irradiance.
For this paper, we normalize the *absolute* radiative effect by the TOA downwelling irradiance
$\left( F_\lambda^{\downarrow,TOA} \right)$ and consider the *relative* radiative effect as percentage of the incident irradiance. Also,
we use spectrally resolved rather than broadband quantities, indicated by subscript λ.

9       The TOA shortwave CRE is always negative (*cooling* effect) because the reflected

irradiance $F_\lambda^{\uparrow,TOA}$ in presence of clouds is larger than for clear-sky conditions. The surface
shortwave CRE is also negative because clouds decrease the transmitted irradiance $F_\lambda^{\downarrow,SUR}$, at
least for homogeneous conditions; broken clouds can locally increase surface insolation. In
contrast to the shortwave CRE at TOA and at the surface, homogeneous clouds have a *warming*
effect on the layer in which they reside. This can be quantified in terms of the layer property
absorptance

$$A_\lambda = \left[ \frac{F_\lambda^{\downarrow,top} - F_\lambda^{\uparrow,top}}{F_\lambda^{\downarrow,top}} - \frac{F_\lambda^{\downarrow,base} - F_\lambda^{\uparrow,base}}{F_\lambda^{\downarrow,top}} \right] \times 100\% \tag{3}$$

for a cloud located between $h_{top}$ and $h_{base}$ with the same normalization as used above for the
relative CRE. It can be determined from aircraft measurements by collocated legs above and
below the cloud (Schmidt et al., 2010). The warming within the layer arises from absorption ($A >$
0) primarily in the near-infrared wavelength range (1 μm < λ < 4 μm). Similarly, as absorptance,
layer transmittance and reflectance are defined as

$$T_\lambda = \left( \frac{F_\lambda^{\downarrow,base}}{F_\lambda^{\downarrow,top}} \right) \times 100\% \tag{4}$$



and $R_\lambda = \left( \dfrac{F_\lambda^{\uparrow,top} - F_\lambda^{\uparrow,base}}{F_\lambda^{\downarrow,top}} \right) \times 100\%$ . (5)
Related to layer reflectance is the albedo $\alpha_\lambda = F_\lambda^\uparrow / F_\lambda^\downarrow$ (identical to $R_\lambda$ for zero surface albedo).
The sum of layer absorptance, transmittance, and reflectance defined in this way is 100% and
thus satisfies energy conservation for horizontally homogeneous layers. For individual pixel sub-
volumes within an inhomogeneous layer (voxels), $A_\lambda$ in Eq. (3) can be replaced with $A_\lambda + H_\lambda \equiv$
$V_\lambda$ where $V_\lambda$ stands for the vertical flux divergence (the net irradiance difference above and below
a layer). In this way, energy conservation including horizontal transport [Eq. (1)] is retained.
The difference of the CRE at TOA and at the surface from Eq. (2) can be related to Eq. (3)
as follows:

$$CRE^{TOA} - CRE^{surface} = \left[ \frac{\left( F_\lambda^{net,cloud} - F_\lambda^{net,clear} \right)^{TOA}}{F_\lambda^{\downarrow,TOA}} - \frac{\left( F_\lambda^{net,cloud} - F_\lambda^{net,clear} \right)^{surface}}{F_\lambda^{\downarrow,TOA}} \right] \times 100\%$$

10                                                     (6a)

$$= \left[ \frac{\left( F_\lambda^{net,TOA} - F_\lambda^{net,surface} \right)^{cloud}}{F_\lambda^{\downarrow,TOA}} - \frac{\left( F_\lambda^{net,TOA} - F_\lambda^{net,surface} \right)^{clear}}{F_\lambda^{\downarrow,TOA}} \right] \times 100\%$$

11                                                     (6b)

The first term inside the brackets of Eq. (6b) is identical to $A_\lambda$ from Eq. (3) if the boundaries of
the layer $h_{top}$ and $h_{base}$ are extended to the TOA and surface, respectively. We denote this by $\hat{A}_\lambda$
and distinguish full-column properties using a hat ($\hat{A}$, $\hat{H}$, $\hat{R}$, $\hat{T}$) from the layer properties that
bracket only the cloud itself ($A$, $H$, $R$, $T$). The second term in Eq. (6b) stems from "clear-sky"
absorption by atmospheric constituents other than clouds (gases and aerosols). Eq. (6b) can then
be re-written as

$$\hat{A}_\lambda = CRE^{TOA} - CRE^{surface} + \left[ \frac{\left( F_\lambda^{net,TOA} - F_\lambda^{net,surface} \right)^{clear}}{F_\lambda^{\downarrow,TOA}} \right] \times 100\%$$

18                                                     (6c)

which simply means that the total atmospheric column absorption comprises contributions from





the cloud itself as well as from clear-sky absorption.[5] In presence of horizontal inhomogeneities,
the left and right side of Eq. (6c) may be inconsistent unless $\hat{A}_\lambda$ is replaced with $\hat{V}_\lambda = \hat{A}_\lambda + \hat{H}_\lambda$ as
above.

4        Presented in this way, the central role of absorptance and horizontal transport in linking

the net irradiances above and below a cloud [Eq. (3)], as well as the TOA and surface CRE [Eq.
(6c)], becomes clear. While the global TOA CRE can directly be derived from reflected radiances
(Loeb et al., 2005), for example from the Clouds and the Earth's Radiant Energy System
(CERES) on the Aqua and Terra satellites (Wielicki et al., 1996), the derivation of the surface
CRE also requires the knowledge of atmospheric absorptance or transmittance. In the case of
CERES, the required cloud properties are obtained from retrievals of the accompanying imager,
the Moderate Resolution Imaging Spectroradiometer (MODIS) (Minnis et al., 2011). As stated in
the previous section, this is accomplished through lookup tables which are based on 1D
calculations and therefore do not provide $H$.

14        Recognizing the crucial significance of horizontal photon transport for obtaining an

accurate surface CRE, Barker et al. (2012) and Illingworth et al. (2015) described the ambitious
goal of using 3D radiative transport operationally in the European radiative budget experiment
Earth Clouds, Aerosols and Radiation Explorer (EarthCARE). They tested their algorithm with
A-Train data. As a metric for 3D effects, they employed the commonly used difference between
3D and IPA calculations (e.g., Scheirer and Macke, 2003). In a similar manner, Ham et al. (2014)
calculated the effect of horizontal photon transport on cloud absorption, transmission, and
reflected radiance. They found these three quantities to be correlated when stratifying their results
by cloud type after spatial aggregation to at least 5 km.

23        Since the studies cited above pertained to EarthCARE and CERES, they only considered

broadband effects. This does not allow distinguishing between $A_\lambda$ and $H_\lambda$ by means of their

---

[5] The cloud contribution term $CRE^{TOA} - CRE^{surface}$ also contains multiple scattering enhancements

of gas absorption due to clouds (Kindel et al., 2011), which may lead to a considerable increase

of the gas absorption (Schmidt and Pilewski, 2012).





distinct spectral characteristics. Our approach, first presented by Schmidt et al. (2014), bridges
this gap. In this paper, we will focus exclusively on the near-ultraviolet and visible wavelength
range and explore the spectral fingerprint from cloud inhomogeneities in conjunction with
molecular scattering in $H_\lambda$, which also imprints itself on reflected radiances (Song et al., 2015).
We chose to not include aerosols in either study, primarily to isolate the spectral signature of
heterogeneous clouds before considering the more general case of clouds and aerosols in
combination.

8       The spectral dependence of the horizontal photon transport across the full shortwave

range will be published separately (Song, 2016). Our expectation for the future, discussed in the
conclusions (Section 9), is that future energy budget studies will capitalize on the spectral
fingerprint of cloud inhomogeneities and derive $H$ by way of the associated spectral radiance
perturbations.
**3. Cloud Data**

14       Our study builds upon the results by Schmidt et al. (2010) and therefore uses the same

cloud case, a tropical convective core with anvil outflow, observed during the TC[4] experiment on
17 July 2007 (from 1519 to 1535 UTC) by the NASA ER-2 aircraft about 300 km south of
Panama. Two realizations of the observed cloud field were used as input to 3D radiative transfer
calculations, one based on airborne imagery only (as in the earlier study, Section 3.1), and one
based on merged airborne and geostationary imagery (Section 3.2) to study large-scale effects.
**3.1 Sub-scene from ER-2 passive and active remote sensors**

21       Level-2 cloud retrievals of the Moderate Resolution Imaging Spectrometer (MODIS)

Airborne Simulator (MAS: King et al., 1996; King et al., 2010) were combined with reflectivity
profiles from the Cloud Radar System (CRS: Li et al., 2004) as described in detail by Schmidt et
al. (2010). The primary information originates from MAS optical thickness, thermodynamic
phase, effective radius, and cloud top height retrievals for each pixel (x,y) within the imager's
swath (roughly 20 km for a cloud top height of 10 km). The imagery-derived information was
extended into the vertical dimension z by simple approximations:





(1) The effective radius from MAS, $r_e(x,y)$, was used throughout the vertical dimension $z$
although representative only of the topmost layer. Since the study is limited to the near-
ultraviolet and visible wavelength range where cloud absorption is negligible, this
simplification only affects the scattering phase function. Approximating it with that at
cloud top is acceptable because to first approximation, 3D radiative transfer is determined
by the distribution of cloud extinction.
(2) The MAS retrieved optical thickness $\tau(x,y)$ for each pixel was vertically distributed by
using the water content profile from CRS: $WC(z) = 0.137 \times Z^{0.64}$ (Liu and Illingworth,
2000) where $Z$ is the radar reflectivity from CRS in dBZ. Since $WC(z)$ is only available
along the flight track, nadir-only CRS profiles were also used across the entire MAS
swath (shifted vertically by $z_0$ to match the MAS cloud top height at off-nadir pixels).
Cloud extinction $\beta$ for each voxel $(x,y,z)$ was thus obtained as

$$\beta(x,y,z) = \tau_{MAS}(x,y) \times WC(z+z_0) / \sum_z WC(z) .$$

Along the flight track, the mismatch between MAS- and CRS-retrieved cloud top height
is $\leqslant$ 0.5 km. The CRS-derived average cloud top height is 10.8 km, and the mean
geometrical thickness is 3.3 km.
The resulting cloud field was gridded to a resolution of 0.5 km horizontally and 1.0 km vertically
(chosen larger than the mismatch between CRS and MAS in cloud top height).
Figure 1 shows the cloud optical thickness field from MAS after regridding, with the
nadir track highlighted as a dashed line. The length of this scene is 192 km (384 pixels in $x$), and
the width is 17.5 km (35 pixels in $y$).
**3.2 Large-scale field from ER-2 data merged with geostationary imagery**
To generalize our findings to larger scales than 17.5 km, we embedded the sub-scene from
the ER-2 remote sensors in the context of the large-scale cloud field as retrieved from the
Geostationary Operational Environmental Satellite West (GOES-11). The imager onboard
GOES-11 has five channels centered at 0.65, 3.9, 6.7, 10.7 and 12.0 μm. In the sampling region,
cloud property retrievals were produced at 1515 and 1545 UTC (Walther and Heidinger, 2012),



of which we chose the earlier one because it was more consistent with the MAS retrieval.
Figure 2 shows the extended cloud scene (240 km × 240 km). Outside the MAS swath,
GOES-11 retrievals were used instead of those from MAS. Similarly, as for the sub-scene cloud,
the effective radius retrieval was extended throughout the vertical dimension. The optical
thickness was distributed vertically using the CRS profile with the closest match in column-
integrated water path (as compared to the retrieved value from GOES) and adjusted in altitude to
match the cloud top height retrievals from GOES-11. This approach for distributing profile
information from active instrumentation across the swath of a passive imager is more simplistic
than that developed by Barker et al. (2011) who used multi-spectral radiances from MODIS.
Transferring radar information to off-nadir pixels as far away as 120 km is not necessarily
justified due to spatial de-correlation of cloud systems (Miller et al., 2014). However, in the
absence of any other information, it was considered the best alternative to estimating the cloud
vertical structure without any *a priori* knowledge.
**4. Model calculations**
The calculations in this study were performed with the 3D Monte Carlo Atmospheric
Radiative Transfer Simulator (MCARaTS: Iwabuchi, 2006). MCARaTS is an open-source code
written in FORTRAN-90, which can be obtained at sites.google.com/site/mcarats/. It calculates
shortwave and longwave spectral or broadband radiances and irradiances based on a forward
propagating photon transport algorithm. It is optimized to run efficiently on parallel computers.
In addition to the two 3D cloud fields described in Section 3, the standard tropical
summer atmosphere as distributed within the libRadtran radiative transfer package
(www.libradtran.org: Mayer and Kylling, 2005) was used to prescribe the vertical profile of
temperature, pressure, water vapor and other atmospheric gases. For gas molecular scattering, we
calculated the optical thickness for each layer with the approximation by Bodhaine et al. (1999)
and selected the Rayleigh scattering phase function from MCARaTS. For gas molecular
absorption, we adopted the correlated *k*-distribution method described by Coddington et al.
(2008). It was originally based on Mlawer and Clough (1997), modified for the shortwave by
Bergstrom et al. (2003), and was specifically developed for the Solar Spectral Flux Radiometer





(SSFR: Pilewskie et al., 2003). The SSFR instrument line shape (6-8 nm full-width half-
maximum) defines the width of the channels in this study (narrower than MODIS or MAS
channels). The spectrum by Kurucz (1992) served as the extraterrestrial solar spectrum.

4          Calculations were performed at eleven wavelengths ranging from the near ultraviolet to

the very-near infrared (350, 400, 450, 500, 550, 600, 650, 700, 750, 800, 1000 nm) to capture the
spectral dependence of horizontal photon transport over a wide range of molecular scattering. At
1000 nm, molecular scattering is negligible and water vapor absorption is small; cloud absorption
is negligible for all wavelengths. For pixels dominated by ice clouds, the scattering phase
function and single scattering albedo were used from the general habit mixture of the ice cloud
bulk models developed by Baum et al. (2011) (parameterized by the effective radius). For liquid
water clouds (minority of cloud pixels), single scattering albedo and asymmetry parameter from
Mie calculations were used in conjunction with a Henyey-Greenstein phase function (which
generally simplifies irradiance calculations). In this study, all calculations were performed for an
ocean surface albedo (Coddington et al., 2010) and for a solar zenith angle of 35° for consistency
with the earlier publication by Schmidt et al. (2010). The solar azimuth angle was 60° (northeast).
This will be generalized in future work. For each wavelength, $10^{11}$ ($10^{12}$) photons were used for
the sub-scene (large-scale) cloud field, respectively. MCARaTS was run in the forward irradiance
mode with periodic boundary conditions. For each 3D model run, calculations were also
performed using the independent pixel approximation (IPA) where horizontal photon transport is
deactivated.
**5. Relationship between cloud spatial structure, net horizontal photon transport, and its**
**spectral dependence**

23          This section discusses the relationship between spatial structure and spectrally dependent

horizontal photon transport based on the small sub-scene. Since true absorption, $A_\lambda$, is negligible,
$H_\lambda$ is equal to $V_\lambda$, the vertical flux divergence of an inhomogeneous cloud layer as defined in
Section 2, with $h_{top} \approx 13$ km and $h_{base} \approx 8$ km.

27          Table 1 shows the optical thickness and effective radius for the eight highlighted pixels

from Fig. 1 along with $H_0$, the horizontal photon transport at $\lambda = 500$ nm, expressed in percent of



the incident irradiance. Positive values of $H_0$ are related to net photon loss to other pixels
("radiation donors"), negative values to net photon gain ("radiation recipient" pixels). In the small
domain, values as high as 50% and as low as –125% were attained. $H_0$ cannot exceed 100%, but
may go below –100%, in which case the radiation received through the sides of a column or
voxel exceeds that from the top of the domain. Table 1 is sorted by $H_0$ rather than by optical
thickness. It shows immediately that there is no relationship between the optical thickness (or
cloud reflectance) and horizontal photon transport. For example, pixel #6 is a "radiation donor",
whereas pixel #4 with roughly the same optical thickness is a recipient. For the extreme case of
zero cloud optical thickness, the effect of horizontal photon transport had previously been
observed as clear-sky radiance enhancement in the vicinity of clouds (Wen et al., 2007; Várnai
and Marshak, 2009). Statistically, this enhancement is a function of the distance of a pixel to the
nearest cloud. However, the horizontal scale of this dependence varies with the spatial context.
Consequently, the distance to a certain cloud element cannot generally be used to parameterize
3D cloud effects for individual pixels, whether cloud-free or cloud-covered. This is illustrated
when considering pixels #4-#8 in the anvil outflow, which have low optical thickness (around 10)
compared to the convective core (optical thickness ≥ 40) overflown from 15.45-15.48 UTC. The
small contrasts in optical thickness (reflectance) between the pixels in close proximity tend to
drive the sign of $H_0$ to a greater extent than the exchange of radiation with the (bright) core (for
example, #6→#7, #5→#4, #7→#8, but not #5→#6). On the other hand, pixels #2 and #3 have
relatively low values of $H_0$ although they have the largest optical thickness of all eight pixels.
While still donors, the magnitude of net horizontal flux to other pixels seems to be diminished by
the vicinity to the convective core. Overall, the direction, let alone the magnitude of net
horizontal flux, is difficult to predict from the distribution of optical thickness, emphasizing 3D
effects as a non-local phenomenon.
For the highlighted pixels in Table 1 (#5-#8), Figs. 3a shows the spectral shape of $H_\lambda$. The
absolute value $H_\lambda$ increases with wavelength until it reaches an asymptotic value towards near-
infrared wavelengths, which we denote $\Delta_\infty$. Donor pixels ($H_\lambda > 0$) are associated with a positive
spectral slope, $S_\lambda \equiv dH_\lambda/d\lambda > 0$; recipient pixels have a negative spectral slope. Remote sensing



studies (e.g., Marshak et al., 2008; Várnai and Marshak, 2009) had previously established that the
above mentioned *radiance enhancement* for clear-sky pixels near clouds was associated with
"apparent bluing", and proposed molecular scattering as the underlying cause for this spectral
dependence. To demonstrate that the same effect is at work here, molecular scattering was
deactivated in MCARaTS, keeping everything else the same in the calculations. In the resulting
spectra (* symbols in Fig. 3a), the wavelength dependence in the near-ultraviolet and visible
range disappears almost entirely, suggesting molecular scattering as the primary cause for the
spectral shape not only for clear-sky, but also for cloudy pixels. This begs the question (addressed
in the next section) of how it is possible to observe such a significant spectral effect for cloudy
pixels, given that cloud scattering outweighs molecular scattering by far. After turning molecular
scattering off, the remaining variability in $H_\lambda$ is due to the weak dependence of cloud scattering
properties on wavelength and droplet or crystal effective radius, as well as minor gas absorption
features.
To first order, the spectral shape over the range of 350 to 650 nm can be characterized by
a single number—the spectral slope at $\lambda = 500$ nm, $S_0$ (obtained from a linear fit to $H_{\lambda=350-600}$
$_{nm}$). Table 1 lists the value of $S_0$ for the eight pixels from Fig. 1, whereas Fig. 3b depicts the
relationship between $H_0$ and $S_0$ for *every* pixel. It shows that not only the sign, but also the
magnitude of the net horizontal photon transport, is surprisingly well correlated with its slope at
500 nm (in %/100 nm). This suggests that the phenomenon observed by Schmidt et al. (2010) for
a few isolated data points is a general occurrence throughout a heterogeneous cloud field. The
close relationship between the magnitude and spectral shape of net horizontal photon transport is
the basis for the spectral parameterization of $H_\lambda$, developed in the next section.
In $H_0$–$S_0$ space, all IPA calculations (red dots in Fig. 3b) are reduced to the origin because
they do not allow pixel-to-pixel radiation exchange by definition. Owing to periodic boundary
conditions, the cloud domain average of $H$ is zero. The calculations without molecular scattering
(grey dots) confirm that molecular scattering dominates the spectral shape throughout the domain.
The vertical spread of the grey data points is due to the other factors mentioned above (e.g.,
variability in cloud microphysics). To some extent, it is also apparent in the IPA calculations.





**6. Physical mechanism and parameterization**
Our interpretation of Fig. 3 is that $H_\lambda$ can be understood as the combination of two terms:
$H_\lambda = H_\infty + \delta(\lambda)$.                    (7)
1.  The constant offset $H_\infty$ is caused by column-to-column radiation exchange between cloud

elements. This is illustrated by Fig. 4 that shows the vertical profile of (a) downwelling, (b)

net, and (c) upwelling irradiance at 1000 nm wavelength for the cloud field from Fig. 1. A

change of net irradiance between altitudes $z_0$ and $z_1$ corresponds to net radiation loss or gain

within that layer. In this case, the domain-averaged profile of net irradiance (black line in

Fig. 4b) decreases slightly near the surface, due to small absorption in the wing of the 936

10       nm water vapor band[6]. When subsampling over columns with a cloud optical thickness $\tau < 1$,

or $\tau > 120$, the 3D calculations differ from the IPA calculations because column-to-column

radiation transfer is enabled. Above the cloud field, columns with high cloud optical

thickness have higher reflectance than the domain average (Fig. 4c) and collectively lose

radiation to those with lower optical thickness; the opposite is true below the cloud where

columns with high optical thickness have lower transmittance (Fig. 4a). The magnitude of

the net horizontal photon transport (the difference of net irradiances at the bottom and top

altitude of a layer) increases with the geometrical layer thickness. Fig. 5 conceptually depicts

the processes at work. Above clouds, net horizontal photon transport (reflected radiance,

projected into a horizontal plane) occurs from the high- to low-reflectance column. Below

clouds, the direction is reversed because the transmittance of thin clouds is larger than that of

thicker clouds.[7] This simplified figure should *not* be interpreted to suggest that the net

horizontal transport generally occurs along gradients of cloud optical thickness. As stated

---

[6] Alternative choices would be 860 nm (although with non-negligible molecular scattering) or
1040 nm (with small cloud absorption under certain conditions, see LeBlanc et al. (2015)).

[7] Note that below $\tau \approx 4$, directly transmitted radiation dominates the downwelling irradiance, and
the cloud may not act as a "diffuser" as shown in Fig. 5. The direction of the green arrows is then
along the direct beam.





above, its direction and magnitude depends not only on directly adjacent columns, but also
on the large-scale context, which is why a parameterization of 3D cloud effects in clear-sky
areas in terms of the distance to the nearest cloud is only possible in a statistical way, but not
on an individual pixel basis (Wen et al., 2007). The value of $H_\infty$ can be obtained from $H_\lambda$ for
wavelengths where molecular scattering becomes negligible and where cloud and gas
absorption are small compared to $H_\lambda$: $A_\lambda \ll H_\lambda$. For the purpose of this study, we chose $\lambda =$
1000 nm: $H_\infty \approx H_{\lambda=1000\text{ nm}}$.
2.   The spectral perturbation $\delta_\lambda$, superimposed on $H_\infty$, introduces the wavelength dependence of

$H_\lambda$. It is perhaps not immediately intuitive why molecular scattering would reduce the

magnitude of $H_\lambda$ as indicated by the symbolic blue arrows in Fig. 5. Molecular scattering

essentially reduces the directionality of horizontal photon transport by redistributing

radiation, part of which can then be detected as enhanced clear-sky reflectance of clouds

(Marshak et al., 2008). A different, secondary process occurs when radiation is scattered out

of the direct beam in clear-sky areas into cloud shadows (dashed blue arrow in Fig. 5). It is

spectrally dependent as $\delta_\lambda$ but, unlike $\delta_\lambda$, *independent* of $H_\infty$ and its direction—thus

increasing the net radiation under both optically thick and thin clouds. For 550 nm

wavelength and shorter (not shown in Figure 4), the net irradiance does indeed increase

towards the surface, both for $\tau > 120$ and for $\tau < 1$. This secondary effect is not explicitly

captured by the first-order parameterization given below.

We express the proportionality of $\delta_\lambda$ to $H_\infty$ as

$$\delta(\lambda) = -\varepsilon \left( \frac{\lambda}{\lambda_0} \right)^{-x} H_\infty \quad (\varepsilon \geq 0, \lambda_0 = 500 \text{ nm}),\qquad\qquad(8)$$
where $(\lambda/\lambda_0)^{-x}$ describes the wavelength dependence, and $\varepsilon$ is the constant of proportionality.
The layer thickness for which $H_\lambda$ is derived affects both $H_\infty$ and $\delta_\lambda$, but only marginally changes
the correlation *between* them. Therefore, $\varepsilon$ is a general parameter that can be used for relating
spatial inhomogeneities and spectral signature of a cloud scene as a whole. It depends on scene
parameters such as surface albedo, solar zenith angle, and cloud micro- and macrophysics
(including vertical structure). This dependence and the secondary effect due to molecular





scattering mentioned above will be explored in a follow-on publication (Song, 2016). Using Eq.
(8), the spectral slope $S_0$ from the previous section can then be derived as

$$S_0 = \frac{dH_\lambda}{d\lambda}\bigg|_{\lambda=\lambda_0} = \frac{d\delta(\lambda)}{d\lambda}\bigg|_{\lambda=\lambda_0} = x\varepsilon\frac{H_\infty}{\lambda_0}, \tag{9}$$

4          By combining Eqs. (7) and (8), one obtains $H_0 = H_{\lambda=500\ nm} = H_\infty(1 - \varepsilon)$, and Eq. (9) can

be rewritten as

$$S_0 = \frac{x\varepsilon}{1-\varepsilon}\frac{H_0}{\lambda_0}, \tag{10}$$

where $x\varepsilon/(1 - \varepsilon)\lambda_0$ is the slope of the linear regression derived using all pixels in the cloud
domain (for example, Fig. 3b). Alternatively, one can derive both $\varepsilon$ and $x$ for each individual
pixel from the regression of

$$\log\left(-\frac{\delta(\lambda)}{H_\infty}\right) = \log\varepsilon - x\log\frac{\lambda}{\lambda_0} \tag{11}$$

with $\log\varepsilon$ as the intercept and $x$ as the slope, as shown in Fig. 6a. In this example, the fit
parameter is about 4 as would be expected if molecular scattering is the underlying physical
mechanism. The two-dimensional PDF $p(x,\varepsilon)$ for the population of pixels in the domain peaks at
$\{x,\varepsilon\} \approx \{3.85, 0.065\}$ but has a considerable spread in both parameters, which is caused by
pixels with negligible horizontal photon transport (and consequently large uncertainties in the fit
parameters). The dashed lines in Fig. 3a show the fitted spectra (labeled "theoretical") from this
approach. For practical purposes, we fix $x \equiv 4$ for the remainder of this paper. This allows using

$$H_\lambda = H_\infty\left(1 - \varepsilon\left(\frac{\lambda}{\lambda_0}\right)^{-4}\right) \tag{12}$$

instead of Eq. (11) and derive $\varepsilon$ and $H_\infty$ for each pixel from a linear regression of $H_\lambda$ versus
$(\lambda/\lambda_0)^{-4}$ (i.e., $H_\infty$ is no longer a required input parameter as for the logarithmic regression). With



$\varepsilon$ known, $S_0$ can be calculated from Eq. (9)[8], and a domain-wide "effective" $\varepsilon$ can be derived
from the slope of the regression line of $S_0$ versus $H_\infty$ for all pixels (Eq. (10) with $x = 4$). Fig. 7
shows the distribution of $\varepsilon$ as derived from (12) for all those pixels with $\Delta(\varepsilon) < 5\%$. The median
of this distribution (0.069) is almost identical to the "effective" value of $\varepsilon$ (0.067). The standard
deviation of the distribution is about 0.01. This means that the parameterized correlation between
net horizontal transport and its spectral dependence can be applied to the domain as a whole as
well as for individual pixels; if the spectral shape of $H_\lambda$ is known, one can infer its magnitude
throughout the near-ultraviolet and visible wavelength range. The correlation is robust regardless
of the cloud context of a pixel, which is remarkable given the considerable variability in distance-
based measures of 3D cloud effects (Várnai and Marshak, 2009).

11         Although our study was instigated by aircraft measurements, its findings are also relevant

for satellite-based derivations of cloud radiative effects since the spectral perturbations $\delta_\lambda$
propagate into observed radiances and imprint a spectral signature of $H_\lambda$ (Song et al., 2015). In
this context, it is important to emphasize the fundamental difference between radiance and
irradiance and their observation from space and aircraft, respectively. Radiances are mainly
affected by radiative smoothing and roughening within a cloud layer (e.g., Marshak et al., 2006).
In addition, aircraft measurements also exhibit geometrical smoothing in their power spectra
(Schmidt et al., 2007a), especially when acquired high above a cloud field. For this reason,
radiance-derived cloud albedo products such as from aircraft imagers (Schmidt et al., 2007b;
Kindel et al., 2010) often do not match their measured counterparts. Through our study, we now
understand why this mismatch [Fig. 7 in Kindel et al., 2010) is associated with a spectral
inconsistency in the albedo spectra (Schmidt and Pilewskie, 2012)—it can simply be explained
by the term $\delta_\lambda$ in Eq. (7).

24         In principle, the mean albedo of an inhomogeneous cloud field derived from CERES

---

[8] This is more accurate than derivation of the slope from a linear fit to the spectrum as used for

Fig. 3, which, due to the non-linearity of the spectral dependence, differs from that of the tangent

if finite wavelength intervals are used.



observations should be fairly insensitive to 3D effects because they are folded into empirical
anisotropy models of such scene types.[9] By contrast, surface cloud radiative effects are much less
constrained by direct CERES observations because cloud transmittance has to be derived from
concomitant imagery. This is where biases introduced by $H_\lambda$ are most significant. For the
remainder of this paper, we therefore analyze the significance of $H$ for varying degrees of spatial
aggregation (Section 7), and make the connection to cloud transmittance (Section 8).
**7. Scale dependence and spatial aggregation**
The results presented so far (e.g., in Fig. 3b) are based on calculations at a resolution of
0.5 km. The question is whether the correlation between the magnitude and spectral shape of $H$ is
scale invariant, and to what extent the effect of horizontal photon transport can be mitigated by
spatial aggregation. To answer this question, we successively coarsened the pixel resolution to 15
km, the largest super-pixel contained within the MAS swath (Fig. 1). Figure 8a shows that the
correlation is indeed independent of the spatial aggregation scale and thus pixel size. The
magnitude of $H_0$ decreases with pixel size: it ranges from +6% to –5% at 15 km resolution (close
to CERES for nadir viewing), compared to about ±50% at 1-5 km (resolution of various MODIS
level-2 products). Eq. (1) suggests that neglecting horizontal photon transport will cause biases in
pixel-level products such as cloud transmittance and surface insolation. In the next section, we
will examine to what extent horizontal photon transport translates into 3D-1D transmittance
biases. Here, we use the large cloud scene (Fig. 2) to estimate for which aggregation scale beyond
15 km the magnitude of $H_0$ drops below the radiometric uncertainty of typical space- or ground-
based radiometers (3-5%), at which point 3D cloud effects become insignificant from a radiative
energy budget point-of-view.
The results for the large scene, shown in Fig. 8b, confirm that the correlation is preserved
for scales up to 70 km. However, $H_0$ at 15 km resolution varies from +17% to –13% throughout

---

[9] This is only true if the empirical anisotropy models adequately accomplish the radiance-to-irradiance conversion.





the large scene domain, much more than in the MAS-only domain (+6% to –5%). One
explanation for this larger range is the greater complexity of the large domain, providing a more
extensive sample of cloud variability than the smaller sub-scene. This becomes quite clear when
looking at the spatial distribution of horizontal photon transport: In Fig. 8c, we chose to plot $S_0$
(y-axis in Fig. 8b) rather than $H_0$. They are practically interchangeable thanks to the correlation
between the two. The distribution of effective donor, recipient, and neutral regions (red, blue,
green, respectively) bears almost no resemblance to the optical thickness field from Fig. 2. This
demonstrates once again that horizontal photon transport cannot be derived from the spatial
distribution of clouds in any simple way; strong contrasts between negative and positive $H_0$ (or
$S_0$) can arise in optically thin boundary layer clouds (southwest corner of Fig. 2 and 8c) as well as
in optically thick areas (deep convection, northeast corner of cloud scene). Extracting the GOES-
MAS large-scene results within the boundaries of the small MAS-only scene (marked by the
green rectangle in Fig. 8c) allows estimating the large-scale exchange of the small domain with
its context. The average value of $H_0$ within the small-scene subset is +7.9%, which means that the
small scene effectively loses photons to its surroundings. This would not be detectable for such a
large aggregation scale (where the entire MAS domain represents a single "super-pixel"). This
net energy export is not reproduced by the calculations based on the MAS-only domain where the
mean value of $H_0$ is zero, in keeping with energy conservation (satisfied by periodic boundary
conditions in the radiative transfer model). The range of $H_0$ in the MAS-only sub-scene of the
GOES-MAS scene is +17% to –6% at 15 km aggregation scale. This is still a larger range than
obtained from the MAS-only calculations (+6% to –5%), even after sub-setting the results from
the large scene to the boundaries of the small ones. The reason is simply that 15 km super-pixel
size is already half the width of the MAS-only domain. Boundary conditions enforce the
convergence of $H_0$ to zero as the area ratio of pixel to domain size approaches 1, which causes an
underestimation of the variability of $H_0$ for large aggregation scales. By contrast, photons can
also travel outside the confines of the domain in the real world as represented by the larger
GOES-MAS cloud scene in our study.

28            This is illustrated in Figure 8d, which shows the range of $H_0$ for both the large and the



small cloud scene as a function of aggregation scale. At small scales, the range is comparable for
the small and large scene. At 15 km aggregation scale, the range obtained from the small scene
has decreased to about half that of the large one. At 50 km pixel resolution, $H_0$ ranges from +7%
to −3% (+5% to −1% at 70 km). It is likely that the boundary conditions imposed on the large
domain also cause an underestimation of the $H_0$ variability at these large scales. Nevertheless,
these results suggest that above 60 km super-pixel size (about 3 × 3 CERES nadir footprints),
horizontal photon transport can be neglected for this cloud scene, based on a 3% uncertainty
threshold. This is only true when aggregating all native-resolution pixels, regardless of whether
they are flagged as clear sky or as cloud-covered. However, sampling cloudy and clear pixels
*separately* would result in much larger biases than 3% because high optical thickness pixels are
more likely to be effective photon donors than low-optical thickness or clear pixels, causing an
asymmetry in the distribution of $H_0$ (Song et al., 2015).
**8. Significance for Cloud Radiative Effect**

14         In this section, we evaluate the ramifications of net horizontal photon transport on

estimates of cloud radiative effects. For any atmospheric column, $H$ is connected to $R$ and $T$
through Eq. (1) and manifests itself in a transmittance and reflectance bias:
$\Delta T = T^{IPA} - T^{3D}$                                                               (13a)
$\Delta R = R^{IPA} - R^{3D}$.                                                               (13a)
Juxtaposing energy conservation for a horizontally homogeneous atmosphere ($T^{IPA} + R^{IPA} = 1$)
with Eq. (1) for conservative scattering ($T^{3D} + R^{3D} = 1 - H$) yields the plausible relationship
$H = \Delta T + \Delta R$,                                                                     (14)
which means that the error introduced by horizontal photon transport is partitioned into
transmittance and reflectance bias. Since the bias $\Delta R$ is folded into the empirical radiance-to-
irradiance conversion employed by CERES, we focus on $\Delta T$ in this study.

25         For the eight super-pixels #11–#18 from Fig. 2, Fig. 9a shows the IPA bias $\Delta T$, ranging

from +2% to +14% in the mid-visible. Its spectral dependence is more complicated than the one
shown for $H$ in Fig. 3a, with a less obvious correlation between magnitude and spectral shape.





Nevertheless, Fig. 9b shows a remarkable correlation between $H_0$ and $\Delta T_0$ ($T^{IPA} - T^{3D}$ at 500
nm) for the same aggregation scales as in Fig. 8b. For example, the $H_0$ range of +15% to –10%
translates into +19% to –12% in $\Delta T_0$ for a horizontal resolution of 20 km. Linear regression
between $H_0$ and $\Delta T_0$ suggests that in this case, $H_0$ propagates mainly into $\Delta T_0$, whereas it is
uncorrelated with $\Delta R_0$ for scales below 20 km (Fig. 10).

6       For simplicity, the spectral dependence of $\Delta T$ as shown in Fig. 9a is approximated by

$$\Delta T_\lambda = T_\lambda^{IPA} - T_\lambda^{3D} = \xi_0\big|_{350-600nm} \times (\lambda - \lambda_0) + (T_0^{IPA} - T_0^{3D}) \; ; \lambda_0 = 500 \text{ nm}$$  (15)
where $\xi_0$ is the spectral slope of $T_\lambda^{IPA} - T_\lambda^{3D}$ calculated from the spectrum between 350 and 600
nm. Fig. 9c shows that the spectral slopes of $H$ and $\Delta T$, $S_0$ and $\xi_0$, are correlated despite the more
complicated spectral dependence of $T$ compared to that of $H$ (Fig. 9a). However, there is clearly
no 1:1 relationship as found between $H_0$ and $\Delta T_0$ above. For example, $S_0 = -10\%/100$ nm
corresponds to only $\xi_0 = -6\%/100$ nm. This changes when extending the vertical layer boundaries
(8-13 km so far, bracketing only the cloud layer itself) to the atmosphere reaching from the
ground to cloud top. This distinction is indicated by hats above all quantities. This is slightly
different from the definition of $\hat{T}$ in Section 2 where the upper boundary is the top of
atmosphere, not the top of the cloud. Fig. 9d not only shows a much stronger spectral dependence
of $\Delta\hat{T}$ (surpassing that of $\hat{H}$) compared to that of $\Delta T$ and $H$ in Fig. 9c, but that the correlation is
no longer scale invariant. This means that the vertical bracket for deriving $T$, $R$, and $H$ has to be
chosen with consideration of the vertical location of the cloud layer. By contrast, the correlation
between $H$ and $S$ as discussed in Section 6 is fairly independent of the layer boundaries.

21       For future studies of IPA-3D biases in satellite-derived estimates of surface cloud

radiative effects, Fig. 4b suggests the center of a cloud as upper boundary of the bracket where
$|dF_{net}/dz|$ reaches a domain-wide minimum because 3D effects can be vertically separated into a
transmittance and reflectance part below and above this level, respectively. Moreover, the
correlation between $\Delta T$ and its spectral dependence $\xi_0$ (not shown) can be exploited to detect
IPA-3D biases in ground-based irradiance measurements below cloud fields (Song, 2016). While
our study suggests that horizontal photon transport mainly propagates into transmittance biases,
there is some indication (Fig. 10) that at scales above 20 km, non-zero values of $H_0$ translate into





albedo (reflected irradiance) biases as well. This increasing correlation with scale is probably
associated with the gradual de-correlation between $\hat{S}_0$ and $\hat{\xi}_0$ observed in Fig. 9b. In order to
improve satellite-based estimates of cloud radiative effects, it is important to understand how $H_0$
is partitioned into $\Delta T$ and $\Delta R$ [Eq. (14)] at different aggregation scales. A detailed study would
need to be conducted for different cloud morphologies, sun angles and surface albedos and is left
for the future. Meanwhile, Song et al. (2015) investigate the link between net horizontal transport
in cloud fields and spectral perturbations in reflected *radiance*.
**9. Summary and conclusions**

9         Deriving the radiative effects of inhomogeneous cloud scenes from observations by

satellite, aircraft, or at the surface is often portrayed as an intractable problem because it cannot
be accomplished by isolating a pixel from its spatial context. At the core of the issue is pixel-to-
pixel exchange of radiation, or net horizontal photon transport, which occurs over a range of
scales. The original motivation for this study was to gain a physical understanding of this
phenomenon's spectral dependence in the near-ultraviolet and visible wavelength range, which
had been found in aircraft irradiance observations (Schmidt et al., 2010). We were able to identify
molecular scattering as the underlying mechanism for the spectral dependence using three-
dimensional radiative transfer calculations with cloud imagery and radar observations as input.
When de-activating molecular scattering in the radiative transfer model, the wavelength
dependence disappeared almost entirely in the vertical flux divergence $V$, which comprises net
horizontal flux density $H$ as well as true layer absorption $A$. To simplify the analysis, we limited
our study to conservative scattering by choosing wavelengths with negligible gas or cloud
absorption ($A \approx 0$), and by excluding aerosols. When activated in the model, molecular scattering
manifested itself as a spectral perturbation (more accurately: modulation) $\delta_\lambda$ to an otherwise
*spectrally neutral* horizontal flux density $H_\infty$, which in turn could be traced back to horizontal
exchange of radiation due to spatial inhomogeneity of cloud elements within the domain. Beyond
the original scope of this study, we made a few surprising discoveries:
1.   The spectral perturbation $\delta_\lambda$ is not independent of the spectrally neutral part $H_\infty$ caused by

28        the clouds themselves. Instead, the mid-visible spectral slope of $H_\lambda$ is correlated with $H$ itself





1    (i.e., with the magnitude of the spectrally neutral part $H_\infty$), which led to the simple

2    parameterization

$$\delta_\lambda = -\varepsilon \left( \frac{\lambda}{\lambda_0} \right)^{-x} H_\infty .$$

2.  We were able to show that the exponent $x$ is close to 4, which further confirmed molecular

scattering as the dominating physical mechanism behind the spectral perturbation. The

constant of proportionality, $\varepsilon$, can be regarded as universally valid for all pixels within the

cloud domain, independently of the vertical or horizontal spatial distribution of clouds. This

means that the spectrally dependent horizontal photon transport can be represented as

$$H_\lambda = H_\infty + \delta_\lambda = H_\infty \left( 1 - \varepsilon \left( \frac{\lambda}{\lambda_0} \right)^{-4} \right)$$

for *each pixel* within the domain with $\varepsilon = 0.7 \pm 0.1$. It seems remarkable that one single

value of $\varepsilon$ should suffice to describe the relationship between the magnitude of $H$ (caused by

clouds) and its spectral dependence (imprinted on $H$ by a completely different physical

process, molecular scattering) – especially considering the range of different clouds within

the domain. The correlation holds for each pixel, no matter what its spatial context may be.

Once $\varepsilon$ is established for a given cloud scene, the spectral perturbations associated with

horizontal photon transport can be derived for each pixel if the value of $H$ is known.

Conversely, if the spectral shape of $H$ is known, the value of $H$ can easily be inferred. This

may be especially significant considering that $H$ cannot be directly observed from space. It is

likely that the spectral perturbations will propagate into the observed radiances. Indeed, Song

et al. (2015) found evidence of this connection in aircraft data. In fact, Várnai and Marshak

(2009) previously reported this effect in clear-sky radiance observations near clouds. The

close correlation that we found in our study may be a pathway to inferring the magnitude of

$H$ from its spectral manifestation in the observed radiances.

3.  The correlation and parameterization hold for a range of spatial aggregation scales, and are

fairly independent of the location of the bracketing altitudes that define the layer. This scale





invariance only breaks down when extending a layer very close to the surface where a
secondary spectral effect has to be factored in (see Section 6 and dashed arrow in Figure 5).
4.  The observed correlation between $H$ and its spectral shape can also be found between

4        transmitted irradiance $T$ and its spectral shape, although it is not scale invariant beyond

5        20 km.

5.  $H$ is correlated with $\Delta T$, the IPA bias for each pixel, but not $\Delta R$ (at least at small scales). This

7        means that 3D cloud effects in the form of horizontal photon transport translate almost

8        exclusively into a transmittance bias. At scales above 20 km, a correlation between $H$ and $\Delta R$

9        does emerge, which requires further study. The correlation between $H$ and $\Delta T$ can potentially

10       be exploited for ground-based spectral irradiance observations (Song, 2016).

Few of these findings could be expected at the outset of our research, and they evoke a number of
new questions:
1.  How does the discovered correlation and the constant of proportionality in its

parameterization, $\varepsilon$, depend on scene parameters such as solar zenith and azimuth angle,

surface albedo (magnitude and spectral dependence), and cloud morphology and

microphysics? What "drives" the parameter $\varepsilon$?

2.  Can the spectral perturbations associated with $H$ indeed be detected in reflected radiances,

and can they be used to infer the magnitude of $H$ indirectly?

3.  Can the findings for the near ultraviolet and visible wavelength range be generalized to the

near-infrared wavelength range where clouds and atmospheric gases do absorb?

4.  What are the implications of our findings for estimating aerosol radiative effects (such as

heating rates) in presence of inhomogeneous cloud fields?

5.  Can the method by Ackerman and Cox (1981) to correct for horizontal photon transport in

aircraft measurements of atmospheric absorption by using a visible channel as basis for the

correction of near-infrared absorption be upheld for future measurements, even in its

modified form proposed by Kassianov and Kogan (2002)?

6.  Can $H$ and $\Delta T$ be derived from spectral perturbations in transmitted irradiance observations

by ground-based spectrometers?



Question 2 will be partially addressed by Song et al. (2015); questions 1, 3, 5, and 6 are discussed
by Song (2016) and will be further investigated in future publications. Furthermore, questions 3
and 4 are subject of active research in the framework of an ongoing or planned field missions
(NASA ORACLES and NSF ONFIRE, dedicated to the radiative effects and remote sensing of
aerosol in vicinity to clouds). This publication constitutes a further contribution to the emerging
field of cloud-aerosol spectroscopy (Schmidt and Pilewskie, 2012), which is expected to improve
the estimation of cloud-aerosol parameters and their radiative effects through spectrally resolved
observations from the ground, aircraft, and, ultimately, space.
**Acknowledgements**
The research presented in this paper was supported by NASA grant NNX14AP72G ("Linking the
Radiative Energy Budget and Remote Sensing of Cloud and Aerosol Fields") within the radiation
sciences program. The calculations were performed on the supercomputer "Janus", which is
supported by the National Science Foundation (award number CNS-0821794) and the University
of Colorado Boulder. It is a joint effort of the University of Colorado Boulder, the University of
Colorado Denver, and the National Center for Atmospheric Research. Janus is operated by the
University of Colorado Boulder.



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





1    **Table 1.** Cloud optical thickness $\tau$, effective radius $r_e$, and values of $H_0$ and $S_0$ for the eight pixels

2    highlighted in Fig. 1 (sorted by $H_0$). For pixels 5, 6, 7, 8, Fig. 3a shows the spectral shape of $H_\lambda$.

| Pixel | $\tau$ | $r_e$ ($\mu$m) | $H_0$ (%) | $S_0$ (%/100 nm) |
|---|---|---|---|---|
| 6 | 10.3 | 27.5 | 28.92 | 2.36 |
| 1 | 13.0 | 30.1 | 21.17 | 1.56 |
| 3 | 21.2 | 30.0 | 13.04 | 1.08 |
| 2 | 18.1 | 30.6 | 9.92 | 1.63 |
| 5 | 12.2 | 27.5 | 4.95 | 0.48 |
| 7 | 8.0 | 27.8 | −5.18 | -0.78 |
| 4 | 11.8 | 28.2 | −18.7 | −1.54 |
| 8 | 7.7 | 24.2 | −24.13 | −2.46 |



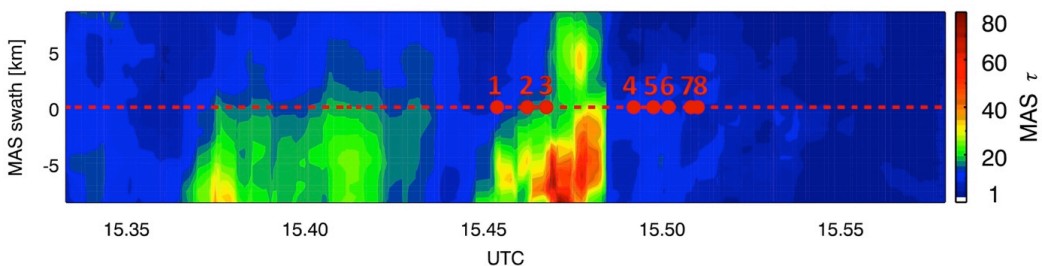

**Fig. 1.** Cloud optical thickness from MAS along an ER-2 leg from 17 July 2007 (length: 192 km,
swath: 17.5 km), regridded to a horizontal resolution of 500 m. The red dashed line indicates the
ER-2 flight track in the center of the MAS swath. Results of net horizontal photon transport for
the eight highlighted pixels are shown in Table 1 and Fig. 3a.





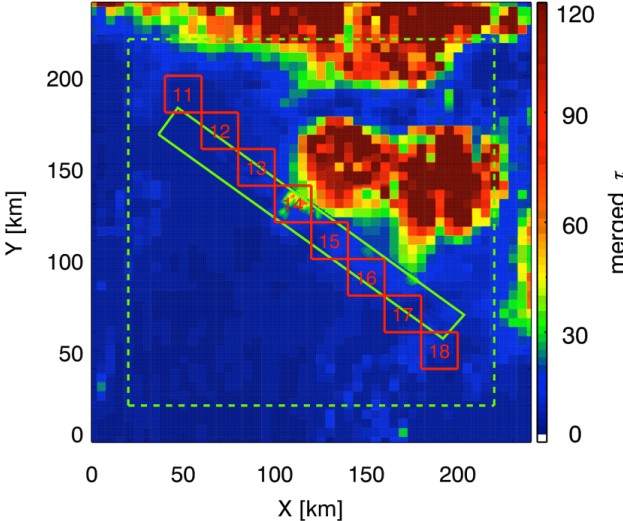

**Fig. 2.** Optical thickness of the large-scale cloud field. The green rectangle marks the embedded
MAS swath (Fig. 1); the red squares mark 20 km "super-pixels" within the scene. Radiative
transfer model output outside the dashed green square is discarded (see Section 7).





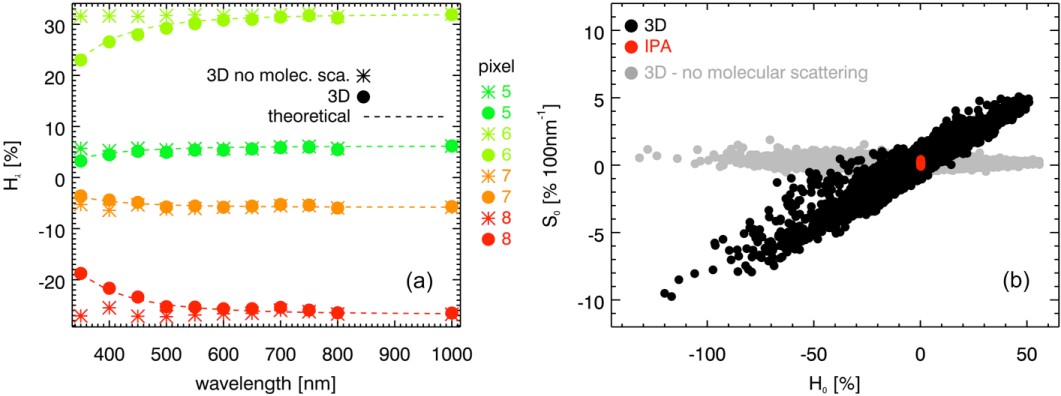

**Fig. 3**. (a) The $H_\lambda$ spectra of pixels {5,6,7,8} from Fig. 1 and Table 1 with (•) and without (∗)
molecular scattering in the 3D calculations, as well as a fit based on Eq. (12) from Section 6
(dashed lines). (b) Spectral slope ($S_0$) vs. net horizontal photon transport ($H_0$) from (a) (both at
500 nm) for all the pixels from Fig. 1. Only 3D calculations with molecular scattering (black
dots) show the systematic correlation between $H_0$ and $S_0$. Disabling molecular scattering (grey
dots) incorrectly predicts a spectrally neutral (flat) $H_\lambda$ ($S_0 \approx 0$ for all pixels). By definition, 1D
calculations (IPA, red dots) do not reproduce net horizontal photon transport at all ($H_0 = 0$ for all
pixels).





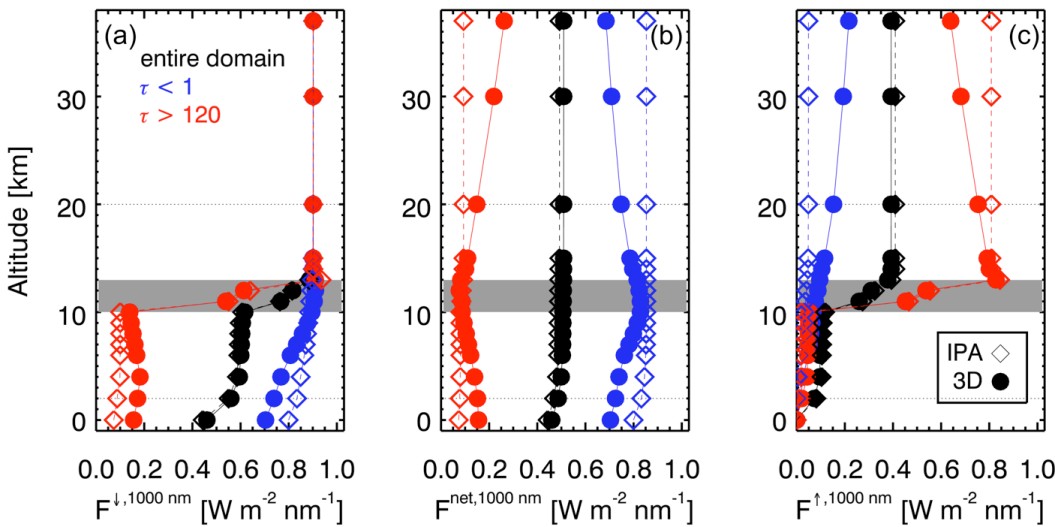

**Fig. 4.** Profiles of (a) downwelling, (b) net, and (c) upwelling irradiance at 1000 nm for the cloud

field from Fig. 1. The location of the cloud layer is marked in grey. Both IPA (dashed line,

hollow symbols) and 3D calculations (solid line, full symbols) are shown, averaged over the full

domain (black), over all columns with $\tau < 1$ (blue) and over columns with $\tau$. 120 (red).



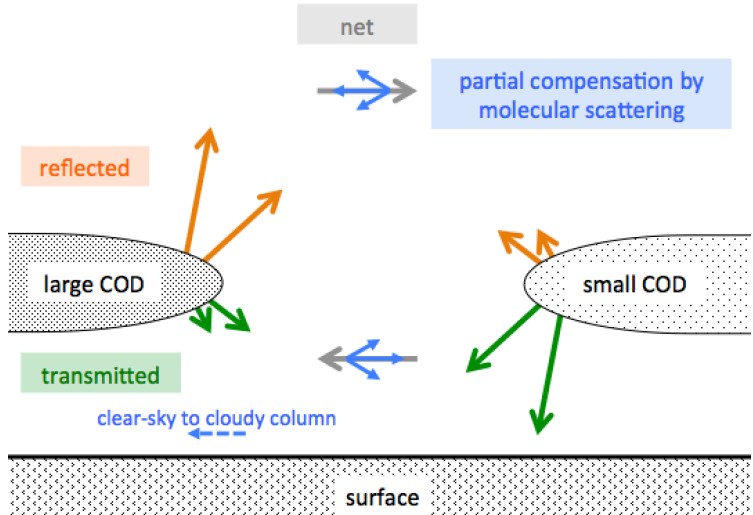

2    **Fig. 5.** Conceptual visualization of the mechanism of horizontal photon transport.




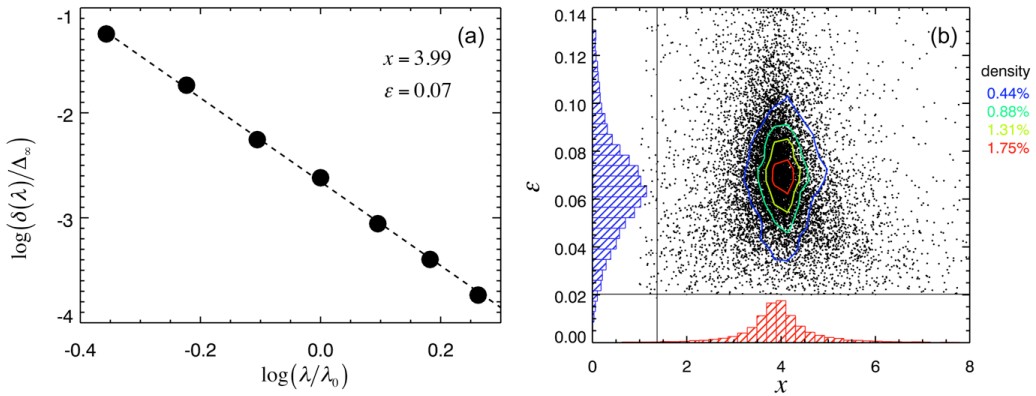

**Fig. 6.** (a) An example of the linear regression between $\log\dfrac{\delta(\lambda)}{H_\infty}$ versus $\log\dfrac{\lambda}{\lambda_0}$, from which the

values of $x$ and $\varepsilon$ can be derived. (b) The scatter plot of $x$ versus $\varepsilon$ for all pixels, joint PDFs $p(x,\varepsilon)$

(contours) as well as the marginal PDFs $p(x)$ and $p(\varepsilon)$ (histograms). The peak of $p(x,\varepsilon)$, and thus

the most likely values $\{x,\varepsilon\}$ values for the cloud field is located at $\{3.85, 0.065\}$, and the domain-

averaged values are $\{3.91, 0.070\}$.





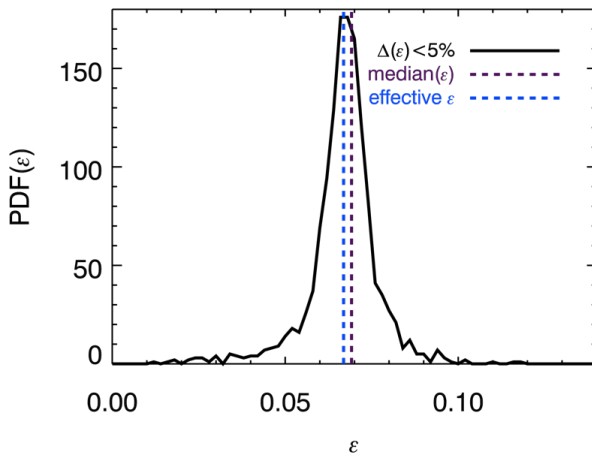

2    **Fig. 7.** PDF of $\varepsilon$ for all pixels with $\Delta(\varepsilon) < 5\%$, median (purple dashed line), and domain-wide

3    *effective* $\varepsilon$ derived from regression of $S_0$ vs. $H_\infty$ (blue dashed line).



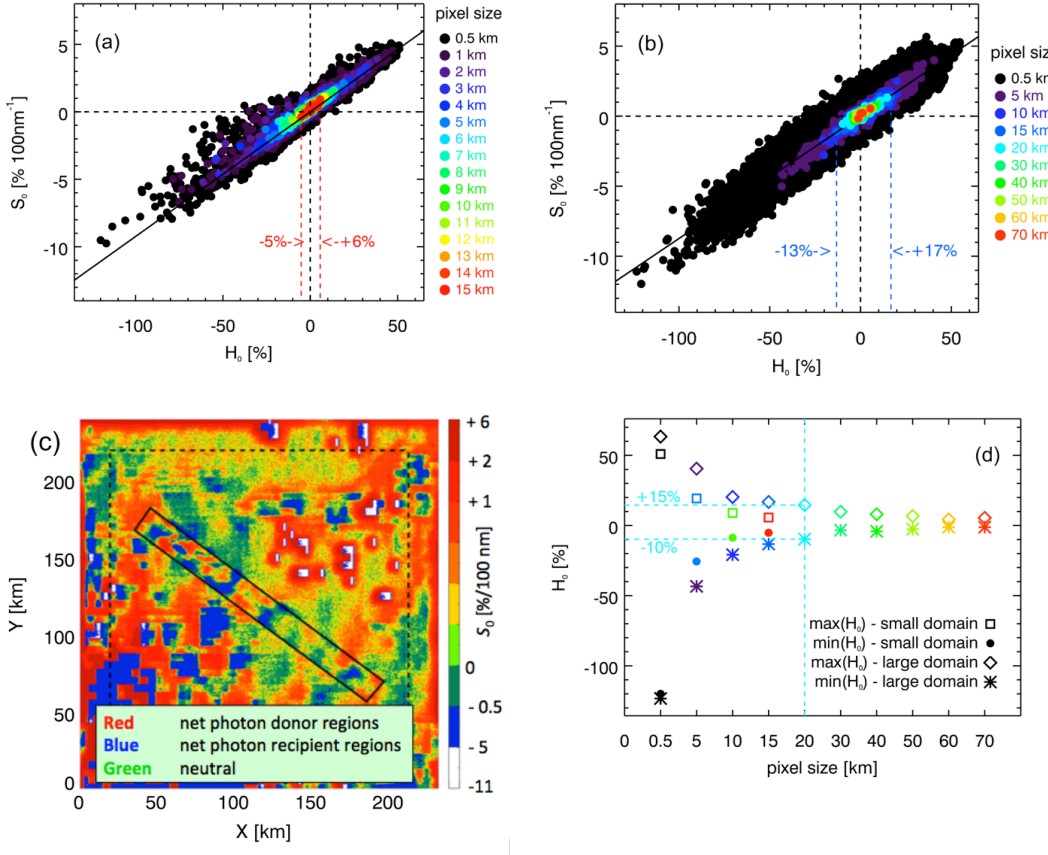

**Fig. 8**. Scatter plot of $S_0$ versus $H_0$ as obtained from linear regression of Eq. (12) for (a) the small
domain from Fig. 1 and (b) the large-scale domain from Fig. 2, spatially aggregated to different
scales, including the 20 km "super pixels" as highlighted in Fig. 2 (red squares). The dashed lines
indicate the range for 15 km pixels. (c) Spatial distribution of $S_0$ from (b). Red (blue) indicates
net photon "donor" ("recipient") pixels, and green "neutral zones" ($H_\lambda \approx S_0 \approx 0$). (d) Dependence
of max($H$) and min($H$) on spatial aggregation scale (km). The color is the same as in (b).





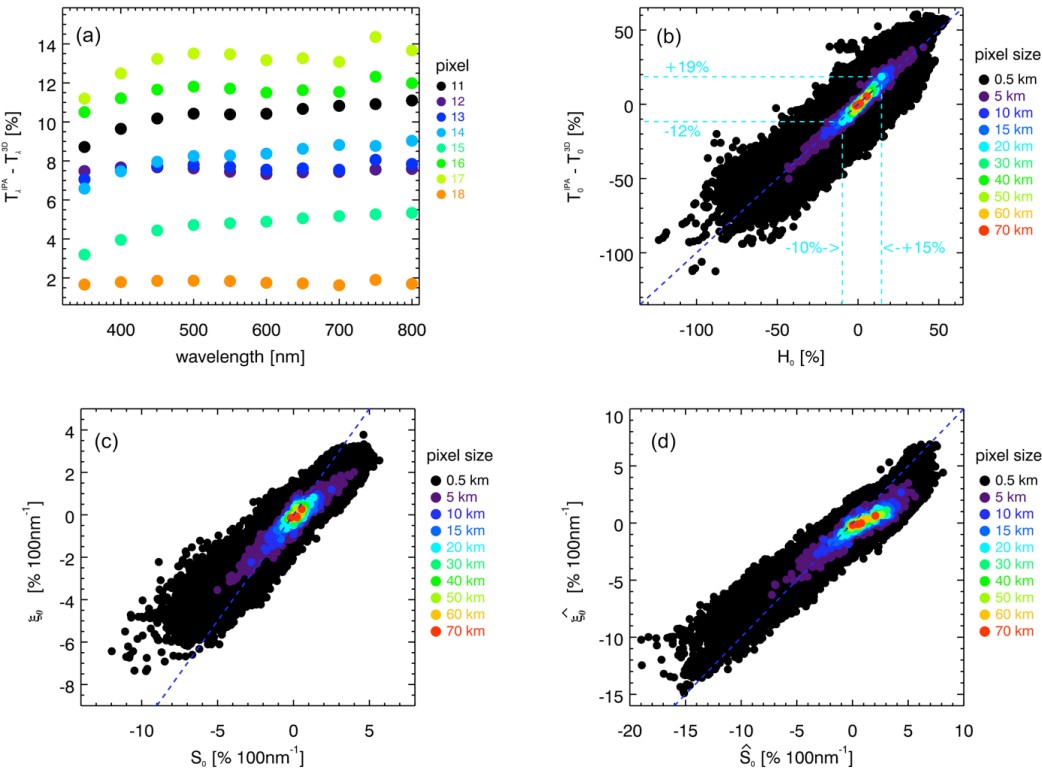

**Fig. 9.** (a) Transmittance biases (IPA-3D transmittance) for the eight super-pixels from Fig. 2. (b) Correlation between net horizontal photon transport from Fig. 8b and transmittance bias for multiple spatial aggregation scales. The dashed lines indicate the range of variability for 20 km super-pixel size. (c) Correlation of the *slopes* of the quantities from (b). (d) Same as (c), but for a bracket from the surface to cloud top, rather than the cloud layer only.





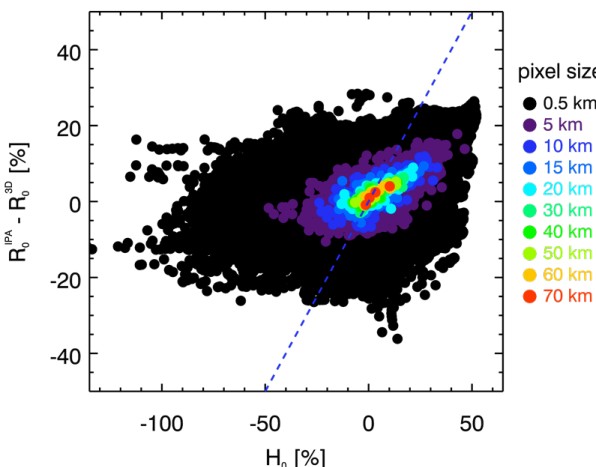

2 **Fig. 10.** $H_0$ is only weakly correlated with reflectance biases $\Delta R_0$ (IPA-3D reflectance) at scales

3 below 15 km, which means that, statistically, biases introduced by horizontal photon transport

4 propagate primarily into transmittance, not albedo. This changes for larger scales.

