# Peer review of "The spectral signature of cloud spatial structure in shortwave"

_Atmospheric Chemistry and Physics, 2015_

## Referee Comment (RC1) · Anonymous Referee #1 · 28 Mar 2016

Overview: This paper shows a robust relationship between net horizontal transport (H) and slope of horizontal transport spectrum ($\partial H\lambda/\partial\lambda$). This relation holds regardless of spatial scales. The authors try to parameterize horizontal net transport as a function of the slope in the spectrum. The authors fully describe radiative terms and equations so that readers are easy to follow them. The figures and tables in this paper clearly show what the authors want to discuss. The reviewer thinks that this manuscript is acceptable to ACP after minor revisions.

General Points #1 Even though this paper contains plentiful new findings and scientific discussions, I feel that the manuscript lacks coherence. I believe that the manuscript can be significantly improved if the authors rearrange paragraphs and shorten unnecessary explanations in Introduction and Discussions.

**2 This manuscript clearly showed a reliable relationship between horizontal net transport and spectral dependency, built a parameterization function, and solved coefficients of the function, such as $\varepsilon$. This is an excellent work indeed. However, it is also important to give a specific direction how the users can apply the parameterization method for inferring 3D effects. I think this is briefly discussed in Section 9 (page 23, line 4-23), so the authors can simply add more detailed explanation/justification of the parameterization in Sections 6 or 9.**

**3 As also commented in the manuscript, the relationship between H and S was inferred in Schmidt et al. (2010). In my understanding, the paper definitely shows new findings, such as a strong linear relationship on a pixel-basis, confirmation of molecular effects from the sensitivity study, and parameterization for the future applications. If this paper highlights new findings in Abstract and Introduction clearly, the readers would catch them more easily.**

Specific points #1 In Abstract, it might be necessary to comment significance of 3D effects, but the authors can simply mention it here and discuss in more detail in later sections. It seems this long discussion hinders main points of this paper (the strong linear relationship that authors found and devise a parameterization method).

**2 Line 1, Page 2: It is not clear what spectral radiance perturbation means. Please explain spectral radiance perturbation, or remove the last sentence of Abstract.**

**3 Line 5-10, Page 3: "The spectral dependence" and the following sentence, I am not sure why the fact - |H| at visible band is similar to |A| at near-infrared - is related to significance of H in broadband A. These two sentences do not seem cause and effect. Please revise them.**

**4 The authors often used footnotes. However, ACP does not recommended footnotes because they disrupt the flow of text. Please consider removing footnotes and includes them in the main text. Please refer to http://www.atmospheric-chemistry-and-physics.net/for_authors/manuscript_preparation.html .**

**5 Line 6-9, Page 4:"In an accompanying paper..." In my understanding, we will need results of Song et al. (2015) to infer dH/dλ from satellite radiance measurements. Once we get dH/dλ from the satellite measurements (or slope), we can estimate H from the parameterization equation in this manuscript. I think this discussion is more relevant when authors explain possible application, e.g. Section 9. It does not carry practical knowledge to readers in Introduction stage.**

**6 Line 8, Page 8: "The spectral dependence of ..the full shortwave range" I think the authors cited Song et al. (2016) since this manuscript considered part of shortwave (< 1000 nm). Please state the wavelength range that this study covers.**

**7 Line 1, Page 10: "we chose the earlier one because it was more consistent with the MAS retrieval" The 1515 UTC is more consistent with MAS in terms of cloud optical depth? Or perhaps 1515 UTC is closer to MAS observation time? Please clarify this.**

**8 Figure 2: From Figure 2, it seems that MAS domain is located boundary of cloud system, according to GOES retrieval. Figure 1 still shows large optical depth up to 80. How consistent MAS and GOES optical depths?**

**9 Line 16, Page 11: It would be helpful if the authors provide # of photons per pixel and corresponding accuracy (e.g. $1/\sqrt{N}$).**

**10 Line 3, Page 12: Is it true that H0 cannot exceed 100%? H0 is divergence of horizontal photon transport (e.g. Eq. A7 in Marshak et al. (1998). Therefore, it should be rare, but isn't it theoretically possible that H0 > 100%?**

Marshak et al. (1998) Biases in Shortwave Column Absorption in the Presence of Fractal Clouds, J CLI, 11, 431-446.

**11 Line 3-4 page 13: molecular scattering as the underlying cause for this spectral dependence. This is a bit different from conclusion in Schmidt et al. (2010) (paragraph [33]). Could the authors explain the difference?**

**12 Line 14-18, Page 17: "In this context, it is....above a cloud field." It is hard to**

[Figure]

understand this paragraph. Could the authors consider revise this paragraph? Also radiance in this paragraph means spectral radiance and irradiance is angle-integrated spectral radiance?

**13 Line 1, page 18: CERES algorithm converts broadband radiance into irradiance without taking into account 3D effects, even though the ADM is based on observation. For example, if the CERES observes radiance in illumination side, radiance for that angle is higher than other angles, but ADM does not consider this. Therefore, I guess the derived irradiance is not completely free from 3D errors. Of course these errors are negligible if we get enough samples and take average spatially and temporally.**

**14 Line 19-20, Page 20 I wonder why two equations in line 19-20 do not absorption terms. TIPA + RIPA + AIPA = 1 T3D + R3D+A3D+H = 1 Then Eq. (14) is H = $\Delta$T +$\Delta$R + $\Delta$A**

**15 From Eq. (14), horizontal transport term H is partitioned into 3D effects on reflection, absorptance, and transmittance ($\Delta$T, $\Delta$R, and $\Delta$A). I think $\Delta$T is strongly correlated with H since absolute magnitude of $\Delta$T is the largest among $\Delta$T, $\Delta$R, and $\Delta$A. Note that cloud albedo is 30%, atmosphere transmittance is 50%, and atmosphere absorption is 20%.**

**16 The authors noted that 3D effects are significant even for large scale. However, previous studies already showed that instantaneous 3D effects might be large, but domain-averaged 3D effects are small. I think the authors need to use 'instantaneous' term if necessary, to differentiate from domain-averaged 3D effects.**

---

## Referee Comment (RC2) · Anonymous Referee #2 · 8 Apr 2016

Review report on: *The spectral signature of cloud spatial structure in shortwave irradiance* by Song et al.

Overview:
The manuscript discusses the spectral net horizontal photon transport in shortwave irradiance fields. Since 3D radiative effects are often discussed in terms of retrieval uncertainties of cloud properties based on radiance measurements, this paper aims on layer properties that are linked to the energy budget.

The authors clearly motivate the relevance of the problem. 3D radiative transfer simulations were applied to determine the magnitude of 3D radiative effects, and to find a reason for the simulated spectral dependence. They found a correlation between magnitude of net horizontal transport and its spectral slope which is parameterized.

The data set and the radiative transfer model is well described. The structure of the manuscript is mainly straight forward. The extensive summary helps the reader significantly to recapitulates the major findings of the work, since there a lot of information given in the main part of the manuscript.

The manuscript is highly recommended for publication in ACP. Nevertheless, a few minor comments should be addressed first.

General comments:

1): There are several places in the manuscript related to radiances instead of irradiances (e.g., p17, l11ff) . For the flow of the paper discussions concerning the relation between $H$ and radiance measurements by satellites should be shifted to the end of the paper.

2): It is not completely clear how to use your findings for other users. How can we improve for example layer properties calculations from airborne irradiance measurements with respect to horizontal photon transport?

Specific comments:

1): In the last sentence of the abstract the authors mention a companion paper. It is not necessary to refer to this publication in the abstract. Rather the authors should give an example how and where the parametrization can be applied for other users.

2): (p3, l7) "can assume similar values as the absorbed irradiance"; Comparing the apparent absorption shown in Fig. 4a (500 nm) and 4b (1600 nm) in Schmidt et al. (2010) I identify the more the same magnitude than similar values. It's still a variable factor between the numbers. Use "same magnitude" instead "similar values". In addition, the authors should give reasons for smaller $H$-values in the NIR.

3): (p3,l20ff) The wavelength dependence of horizontal photon transport is mentioned here. Could you give a more detailed literature review on this since it is crucial for the entire manuscript?

4): (p4, l2-15) The paragraph is a mixture of outline and outlook (l6-9). Please

strengthened the content. A structure of the paper is already described in the last paragraph of the introduction. Therefore the idea of the paper should be presented before (performing 3D and 1D simulations with a measured cloud data set, identifying $H$ and it's spectral behavior, . . .) without prejudging the results.

5): (p5, l18) Eq. (3) states the spectral absorptance. Add here directly, that these layer properties are valid for homogeneous conditions without horizontal photon transport. The reader might be confused otherwise because Eq. (3) contradicts Eq. (1) without this restriction (as noted only on p.6, l5-7).

6): (p8, l8-12) This paragraph gives an outlook. Better put this at the end of the manuscript.

7): (p9, l4-6) As stated by the authors using height-constant effective radii has an effect on the vertical distribution of the phase functions which probably differ from reality. Why does the phase function don't affect the 3D radiative transfer? Changes of the phase function result in changes of the scattering direction. Maybe this is not as relevant as for radiance simulations. Please clarify.

8): (p9, l8) Please define $WC$.

9): (p9, l17) Please justify the choice of spatial resolution (with respect to typical spatial scales of radiative smoothing).

10): (p11, l16) What will be generalized? The solar position?

11): (p12, l8-11) The enhancement of radiance in the vicinity of clouds is mentioned here. Can you cite also papers dealing with the enhancement of irradiances? Add also the fact that this effect is wavelength-dependent.

12): (p13, l15) Could you insert the linear fit in Fig. 3a?

13): (p13, l24) "pixel-to-pixel radiation exchange" $\rightarrow$ Please add "horizontal" here. There is of course a vertical exchange of photons.

14): (p18, l16-19) "Eq. (1) suggests. . ." In my opinion these two sentences do not contribute significantly to the context of this section. Referring to transmittance here somehow interrupts the flow of the discussion on spatial aggregation.

15): (p20, l20) Please motivate the restriction of conservative scattering here, otherwise the missing absorption term might confuse the reader.

16): (Sect. 8, first paragraph) To make sure that the equations are valid only for a specific wavelength range, the index "$\lambda$" would be helpful for $H$, $R$, $T$,...

17): (p23, l4, l10) If you give numbers here then you have to mention that these numbers are case specific with respect to surface albedo and solar position.

18): (Sect. 9) Be more consistent with using indices for $H$. For example p.23, l.16: Is it $H$ or $H_0$ or $H_\lambda$ which has to be known?

19): (Fig3b) Is there any reason for the increasing scattering of 3D-based $S_0 - H_0$ correlation for negative slopes?

Technical comments:

1): Please remove the footnotes.

2): (all Figs.) Check that symbols have italic format.

3): (p12, l25) Figs. $\rightarrow$ Fig. 4): (p13, l25) "$H$" $\rightarrow$ "$H_0$" 5): (p14, l3) "H$_\lambda$" $\rightarrow$ "$H_\lambda$" (italic)
* * *

---

## Author Comment (AC1) · 12 Aug 2016

**Response to review of "The spectral signature of cloud spatial structure in shortwave irradiance" by anonymous Referee #1**

Sebastian Schmidt, corresponding author

We very much appreciate the thorough and positive review of this manuscript and the helpful comments for improving content, clarity, and context within the literature. We are open for further input, should we have mis-interpreted the reviewer's points (point-by-point response below).

Assessment by reviewer: Minor revisions

General points:

**1 Even though this paper contains plentiful new findings and scientific discussions, I feel that the manuscript lacks coherence. I believe that the manuscript can be significantly improved if the authors rearrange paragraphs and shorten unnecessary explanations in Introduction and Discussions.**

We agree with the reviewer and heeded the advice by removing unnecessary explanations (not just in the introduction), especially the ones pertaining to radiances, which interrupted the flow of the paper. It was tempting to allude to this topic in this paper, but we realize that it is better addressed in a companion paper. Rather than going into too much detail here, we instead included a reference to a Ph.D. and the companion paper (Song et al. 2016, to be submitted soon). Changes are highlighted in the revised version of this paper. Most of the changes in response to this comment are in the introduction and in the body of the paper; the Summary & Conclusions section was shortened only slightly because we felt the need to discuss the significance of our findings given the unusually large amount of material covered, and this was appreciated by reviewer #2.

References:

Song, 2016: The Spectral Signature of Cloud Spatial Structure in Shortwave Radiation, *Ph.D. thesis, University of Colorado at Boulder.*

Song, S., K. S. Schmidt, Pilewskie, P., King, M. D., Platnick, S., 2016: Quantifying the spectral signature of heterogeneous clouds in shortwave radiance and irradiance measurements, to be submitted to *JGR SEAC⁴RS special issue*

**2 This manuscript clearly showed a reliable relationship between horizontal net transport and spectral dependency, built a parameterization function, and solved coefficients of the function, such as ε. This is an excellent work indeed. However, it is also important to give a specific direction how the users can apply the parameterization method for inferring 3D effects. I think this is briefly discussed in Section 9 (page 23, line 4-23), so the authors can simply add more detailed explanation/justification of the parameterization in Sections 6 or 9.**

This is a very good point, which was brought up by both reviewers. Indeed, the term "parameterization" might suggest that it can be exploited for inferring, simplifying, or correcting 3D effects, and the authors are currently working on this very topic. However, the parameterization is only the first step towards this goal, and it cannot (yet) be translated into such immediate practical applications, although this is certainly the goal for the future. The purpose of the parameterization is to capture the relationship between net horizontal photon transport and its spectral dependence using one main parameter ($\varepsilon$). The companion paper (Song et al., 2016) will look at the connections between 3D effects on irradiances and radiances. We will include this explanation in the revised version. For example, we conclude the abstract with the following statement: "Since three-dimensional effects depend on the spatial context of a given pixel in a non-trivial way, the spectral dimension of this problem may emerge as the starting point for future bias corrections." In section 6, we included this statement "Although our study was instigated by aircraft measurements, its findings are also relevant for satellite-based derivations of cloud radiative effects since the spectral perturbations $d\lambda$ propagate into observed radiances (Song et al., 2016). This may be exploited in future applications for deriving correction terms for 3D radiative effects via their spectral signature." We hope this clarifies the purpose of the parameterization.

**3 As also commented in the manuscript, the relationship between H and S was inferred in Schmidt et al. (2010). In my understanding, the paper definitely shows new findings, such as a strong linear relationship on a pixel-basis, confirmation of molecular effects from the sensitivity study, and parameterization for the future applications. If this paper highlights new findings in Abstract and Introduction clearly, the readers would catch them more easily.**

We agree – it was somewhat unclear in the abstract what was done in earlier studies vs. this paper. The revised abstract was re-structured significantly, and clearly points out the new aspects of this paper at the very beginning, i.e., identifying the physical mechanism that causes the correlation between spatial structure and spectral signature, as well as the parameterization developed on its basic. The new abstract reads as follows:

"In this paper, we used cloud imagery from a NASA field experiment in conjunction with three-dimensional radiative transfer calculations to show that cloud spatial structure manifests itself as spectral signature in shortwave irradiance fields – specifically in transmittance and net horizontal photon transport in the visible and near-ultraviolet wavelength range. We found a robust correlation between the magnitude of net horizontal photon transport ($H$) and its spectral dependence (slope), which is scale-invariant and holds for the entire pixel population of a domain. This was at first surprising given the large degree of spatial inhomogeneity, but seems to be valid for any cloud field. We prove that the underlying physical mechanism for this phenomenon is molecular scattering in conjunction with cloud inhomogeneity. On this basis, we developed a simple parameterization through a single parameter $\varepsilon$, which quantifies the characteristic spectral signature of spatial heterogeneities. In the case we studied, neglecting net horizontal photon transport leads to a transmittance bias of ±12-19% even at the relatively coarse spatial resolution of 20 kilometers. Since three-dimensional effects depend on the spatial

context of a given pixel in a non-trivial way, the spectral dimension of this problem may emerge as the starting point for future bias corrections."

Specific points:

**1 In Abstract, it might be necessary to comment significance of 3D effects, but the authors can simply mention it here and discuss in more detail in later sections. It seems this long discussion hinders main points of this paper (the strong linear relationship that authors found and devise a parameterization method).**

Agreed; see the point above along with the modified abstract. The discussion of 3D effects for the particular case studied in our paper was moved to the end, to emphasize the main points (presented at the beginning).

**2 Line 1, Page 2: It is not clear what spectral radiance perturbation means. Please explain spectral radiance perturbation, or remove the last sentence of Abstract.**

The last sentence of the abstract was deleted, and a more general statement was added ("Since three-dimensional effects depend on the spatial context of a given pixel in a non-trivial way, the spectral dimension of this problem may emerge as the starting point for future bias corrections.").

**3 Line 5-10, Page 3: "The spectral dependence" and the following sentence, I am not sure why the fact - |H| at visible band is similar to |A| at near-infrared - is related to significance of H in broadband A. These two sentences do not seem cause and effect. Please revise them.**

We revised this section on page 3 to address this problem, it now reads as follows:

"Schmidt et al. (2010) derived *apparent absorption*, the sum of $A$ and $H$, from irradiance measurements aboard the NASA ER-2 and DC-8 aircraft that flew along a collocated path above and below a heterogeneous anvil cloud during the Tropical Composition, Cloud and Climate Coupling Experiment (TC$^4$) (Toon et al., 2010). The results of this study showed that, in absolute terms, $H$ at visible wavelengths (where cloud and gas absorption are negligible) can attain a similar magnitude as the absorbed irradiance $A$ at near-infrared wavelengths. Horizontal photon transport thus has the potential to mimic substantially enhanced absorption. Three-dimensional (3D) calculations confirmed the measurements, and radiative closure was achieved within measurement and model uncertainties without invoking proposed enhanced gas absorption (Arking, 1999) or big cloud droplets (Wiscombe et al., 1984)."

Note that we kept the statement "Horizontal photon transport thus has the potential to mimic substantially enhanced absorption," but removed the term "broadband". What we meant was that a broadband observation of "absorption" by way of collocated legs above

and below a cloud layer is really the wavelength integral of A_lamda + H_lamda, not just A_lamda. If the magnitude of H in the visible is on the same order of magnitude as A in the near-infrared, the contribution of H to the broadband integral of A+H may be comparable to that of A. In fact, it may even outweigh it (not stated in the paper). For this reason, it is important to make spectrally resolved measurements; otherwise it is impossible to separate H and A (in the spirit of the Ackerman & Cox papers).

**4 The authors often used footnotes. However, ACP does not recommended foot- notes because they disrupt the flow of text. Please consider removing footnotes and includes them in the main text. Please refer to http://www.atmospheric-chemistry-and-physics.net/for_authors/manuscript_preparation.html .**

Thank you, all the footnotes were either removed (where not of central importance to the manuscript) or incorporated into the manuscript.

**5 Line 6-9, Page 4:"In an accompanying paper. . ." In my understanding, we will need results of Song et al. (2015) to infer dH/dλ from satellite radiance measurements. Once we get dH/dλ from the satellite measurements (or slope), we can estimate H from the parameterization equation in this manuscript. I think this discussion is more relevant when authors explain possible application, e.g. Section 9. It does not carry practical knowledge to readers in Introduction stage.**

Thank you for catching this; we deleted the radiance-related statement here. The shortened paragraph now reads as follows:

"Further analysis of the relationship between cloud structure and its spectral signature, presented here, revealed a surprisingly robust correlation between the magnitude of H and its spectral slope, dH/dλ. In the course of this paper, we provide evidence for molecular scattering as the physical mechanism behind this correlation and develop a simple parameterization based on this knowledge. We also examine at which spatial aggregation scale H can be ignored and whether the discovered correlation between H and dH/dλ is scale invariant. Finally, we consider the ramifications of our findings on the shortwave surface energy budget and find that while cloud transmittance biases may be significant even after spatial averaging, they are also accompanied by spectral perturbations similar to the ones that we encountered for H. These biases may thus be detectable and correctable using adequate ground-based radiometers."

**6 Line 8, Page 8: "The spectral dependence of ..the full shortwave range" I think the authors cited Song et al. (2016) since this manuscript considered part of shortwave (< 1000 nm). Please state the wavelength range that this study covers.**

The originally cited work (Song 2016, a dissertation, has now been published) actually changed scope and actually no longer covers any wavelengths beyond the near-UV, visible, and very near infrared. We have therefore removed this reference. We would like to point out here that there is work that has been done by Marshak and others for *radiances*, (Marshak, Evans, et al., 2014) and we added a statement to this effect. We also added a reference to Kassianov and Ovtchinnikov (2008).

**7 Line 1, Page 10: "we chose the earlier one because it was more consistent with the MAS retrieval" The 1515 UTC is more consistent with MAS in terms of cloud optical depth? Or perhaps 1515 UTC is closer to MAS observation time? Please clarify this.**

This question allows us to show plots that we chose not to include in the manuscript. As the reviewer suggested, we used cloud optical depth to chose from two possible GOES scenes. The first plot shows the collocated MAS/GOES15:15 optical depth within 0.1° around the ER-2 latitude and longitude along the flight track. The second plot shows the same, but with the later GOES retrieval (15:45).

[Figure]

In terms of the timing, both GOES retrievals would be possible because the ER-2 flight leg (15:21-15:33) is right in between 15:15 and 15:45. However, the comparison of the MAS- and GOES retrieved optical thickness is more consistent when using the 15:15 scene. We changed the text as follows to make this clear: "In the sampling region, cloud property retrievals were produced at 15:15 and 15:45 UTC (Walther and Heidinger, 2012), of which we chose the earlier time because it was more consistent with the MAS retrieval in terms of the optical thickness along the ER-2 track."

**8 Figure 2: From Figure 2, it seems that MAS domain is located boundary of cloud system, according to GOES retrieval. Figure 1 still shows large optical depth up to 80. How consistent MAS and GOES optical depths?**

This observation is correct. The MAS swath does capture the edge of a cloud system (as shown in Figure 2). The color scale of Figures 1 and 2 is different; even GOES shows a

fairly large optical thickness on the NE edge of the MAS swath. Because of the different pixel size, GOES and MAS retrievals are not expected to match exactly. For this reason, the retrievals were aggregated to 0.1° "super-pixels" in the optical thickness plots above. The edge of the cloud system that the reviewer mentions is sampled at UTC=15.47 by the ER-2, and MAS and GOES show optical thickness values of ~20-30 at this aggregation scale. The higher optical thickness values as observed by MAS (~60) are small-scale maxima. In general, GOES and MAS retrievals are consistent within the range of the standard deviation in the 0.1° circle.

Note that the agreement in other retrieval parameters (cloud top height, effective radius) was not as good, in part because of different channel combinations that were used by the MAS / GOES algorithms. We chose not to go into detail about the MAS/GOES consistency in this paper because this is not its main purpose; such studies may be done in a separate paper.

**9 Line 16, Page 11: It would be helpful if the authors provide # of photons per pixel and corresponding accuracy (e.g. 1/sqrt(N)).**

We included some more information on the photon number in the revised manuscript.
Small domain: 1e11 or 7.4e6 per pixel
Large domain: 1e12 or 4.3e6 per pixel
These photon numbers led to sufficiently low noise level. For example, the maximum standard deviation for the upwelling irradiance at the pixel level is 0.008 W/m2/nm at 500 nm.

**10 Line 3, Page 12: Is it true that H0 cannot exceed 100%? H0 is divergence of horizontal photon transport (e.g. Eq. A7 in Marshak et al. (1998). Therefore, it should be rare, but isn't it theoretically possible that H0 > 100%?**

Marshak et al. (1998) Biases in Shortwave Column Absorption in the Presence of Fractal Clouds, J CLI, 11, 431-446.

Thank you for this excellent catch! The reviewer is of course correct; this erroneous statement survived our internal review process. In fact, we found cases (in our own analysis for the next paper) where $H_0$ does exceed 100%. We simply deleted this statement, the revised version reads: "When $H_0$ falls below –100%, the radiation received through the sides of a column or voxel exceeds that from the top of the domain." We don't state that the opposite is also true (for $H_0$>100, but that goes without saying).

**11 Line 3-4 page 13: molecular scattering as the underlying cause for this spectral dependence. This is a bit different from conclusion in Schmidt et al. (2010) (paragraph [33]). Could the authors explain the difference?**

A very good point! We need to provide a little bit of background to explain this. The statement from Schmidt et al. (2010) in question is the following: "Preliminary tests showed that switching off molecular scattering in the RT model did not change the

slope significantly, thus ruling out molecular scattering as the cause for the spectral slope of the apparent absorptance." In light of new evidence, the second part of this statement is, in fact, incorrect, but we must emphasize the word "preliminary". It is true that when switching off molecular scattering, the slope did not change in this earlier study (incidentally done with a different model than used here). We therefore had to assume that the reason for the slope must lie elsewhere. At least two colleagues in the field thought that molecular scattering could not have such a large effect on irradiance (in contrast to radiance where it had been found at this point). While we always suspected molecular scattering, we could not present evidence at this point. In light of this, we should have worded this statement more cautiously. As it only turned out later, the explanation was that the switch in the model was actually inactive (keeping molecular scattering on regardless of the switch settings). We did not suspect this until after the paper was published, at which point we had a conversation with one of the code developers who brought up this possibility. In retrospect, this was a user error because we should have been able to diagnose this problem with further runs. We have since done these tests and found the cause of the problem. The analysis in the current paper (Figure 3) correctly shows that molecular scattering does explain the phenomenon. We added the following statement about the earlier study: "Note that the earlier study by Schmidt et al. (2010) remained inconclusive as to the mechanism of the spectral dependence they observed." This is justified as the earlier study states (in the conclusions): "The physical basis of the spectral shape of near‐UV and visible apparent absorption remains to be explored, as well as the scales over which horizontal photon transport occurs in high‐cloud systems (for example, by embedding the MAS cloud scene in the larger context of GOES retrievals)."

**12 Line 14-18, Page 17: "In this context, it is. . ..above a cloud field." It is hard to understand this paragraph. Could the authors consider revise this paragraph? Also radiance in this paragraph means spectral radiance and irradiance is angle-integrated spectral radiance?**

We simply deleted this paragraph because it distracted from the main content.

**13 Line 1, page 18: CERES algorithm converts broadband radiance into irradiance without taking into account 3D effects, even though the ADM is based on observation. For example, if the CERES observes radiance in illumination side, radiance for that angle is higher than other angles, but ADM does not consider this. Therefore, I guess the derived irradiance is not completely free from 3D errors. Of course these errors are negligible if we get enough samples and take average spatially and temporally.**

We agree, and the Ham et al. (2014) publication (cited in our paper) talks about the effect of horizontal photon transport (not so much about illumination though). However, the 3D errors in transmitted irradiance should be much larger than in albedos because in principle, the ADMs do include spatially inhomogeneous conditions, however sparsely the parameter space may be sampled for those. Also, our statement was meant in the statistical sense, i.e., averaging over multiple

"realizations" of such scene types. We modified our statement as follows: "In principle, the mean albedo of an inhomogeneous cloud field derived from CERES observations should be fairly insensitive to 3D effects because they are statistically folded into anisotropy models of such scene types (if these empirical models adequately accomplish the radiance-to-irradiance conversion for a range of sun-sensor geometries)."

**14 Line 19-20, Page 20 I wonder why two equations in line 19-20 do not [have] absorption terms.**

TIPA+RIPA+AIPA=1

T3D+R3D+A3D+H=1

Then Eq. (14) is H=$\Delta$T+$\Delta$R + $\Delta$A

This set of equations was written for conservative scattering (no absorption), but since the other reviewer also noted the lack of absorption, we made this more clear by slightly rewording as follows: "Juxtaposing energy conservation for a horizontally homogeneous atmosphere ($T$IPA + $R$IPA = 1) with Eq. (1) for conservative scattering (**$A$=0, therefore** $T$3D + $R$3D = 1 – $H$) yields the plausible relationship…"

**15 From Eq. (14), horizontal transport term H is partitioned into 3D effects on reflection, absorptance, and transmittance ($\Delta$T, $\Delta$R, and $\Delta$A). I think $\Delta$T is strongly correlated with H since absolute magnitude of $\Delta$T is the largest among $\Delta$T, $\Delta$R, and $\Delta$A. Note that cloud albedo is 30%, atmosphere transmittance is 50%, and atmosphere absorption is 20%.**

This is an interesting thought, and we believe that this partitioning may need to be investigated in the future. It is indeed plausible that the bias is correlated with the magnitude of T and R itself. However, we did not attempt to do the partitioning in the study and focused mainly on the transmittance in the remainder of the paper. We do note that H at the pixel level is correlated with $\Delta$T, but not with $\Delta$R. However, we do not draw conclusions about the magnitude of the two biases. Comparing Figure 9b with Figure 10 does show that the range of $\Delta$T is larger than that of $\Delta$R, however the point there is not the magnitude but the correlation with H. As to Figure 10, it was surprising to us that R and H do become correlated at scales greater than 5 kilometers.

**16 The authors noted that 3D effects are significant even for large scale. However, previous studies already showed that instantaneous 3D effects might be large, but domain-averaged 3D effects are small. I think the authors need to use 'instantaneous' term if necessary, to differentiate from domain-averaged 3D effects.**

The emphasis of the paper as a whole was on the spectral aspect of this problem, not on the magnitude of the 3D effect, for which the single case presented in the paper

would not have sufficient statistics anyway. We confirm that we mean a *local* 3D effect, rather than the domain-average effect. We prefer "local" to "instantaneous" as suggested by the reviewer because it is tied to space rather than time. Where appropriate, we added "local" in the few occurrences where we do talk about magnitudes. For example, the section in the abstract reads as follows: "In the case we studied, neglecting net horizontal photon transport leads to a **local** transmittance bias of ±12-19% even at the relatively coarse spatial resolution of 20 kilometers." More changes have been made to section 7. In other cases, we made clear that we are talking about pixel-level effects and biases. We agree that in the domain average (as shown in previous papers), 3D biases become small. At the same time though, our study showed that even aggregating the data to large scales, significant biases survive. Figure 8d is meant to illustrate this. One can essentially read off the biases for various aggregation scales. For example, at 0.5 km pixel size, we get >50% biases. Averaging to 20 km decreases the bias to just over 10%, and it eventually disappears at even larger aggregation scales. We do believe that Figure 8d and the text accompanying makes this clear. We fully agree with the comment by the reviewer and do not contradict earlier studies.

More recent research (Song, 2016) shows that by considering 3D effect on irradiance (as done in this paper) *and* on cloud remote sensing may lead to biases in transmitted irradiance estimates that do not disappear with increasing scale but survive averaging. This research will also be presented in a separate paper (Song et al., 2016).

---

## Author Comment (AC2) · 12 Aug 2016

**Response to review of "The spectral signature of cloud spatial structure in shortwave irradiance" by anonymous Referee #2**

Sebastian Schmidt, corresponding author

We thank the reviewer for the positive assessment of the manuscript and the helpful comments regarding the clarity and cohesion. We shortened the introduction, leaving out unnecessary references to the radiance aspect of the problem, which helped the cohesion of the manuscript. Owing to the reviewer's positive feedback, we kept the conclusion section largely unchanged. Regarding the applicability of our parameterization, see our response to general comment #2. [Note that page/line numbers refer to the original, not the revised manuscript.]

Assessment: Minor Revisions

General comments:

**1 There are several places in the manuscript related to radiances instead of irradiances (e.g., p17, l11ff) . For the flow of the paper discussions concerning the relation between H and radiance measurements by satellites should be shifted to the end of the paper.**

This is an excellent point; the other reviewer had a similar comment. The discussion of radiances interrupted the flow of the paper; we removed multiple occurrences in the body of the paper and discussed it mainly in the conclusions. Rather than going into too much detail in this paper, we instead included a reference to a Ph.D. and the companion paper (Song et al. 2016, to be submitted soon). Changes are highlighted in the revised version of this paper.

References:

Song, 2016: The Spectral Signature of Cloud Spatial Structure in Shortwave Radiation, *Ph.D. thesis, University of Colorado at Boulder.*

Song, S., K. S. Schmidt, Pilewskie, P., King, M. D., Platnick, S., 2016: Quantifying the spectral signature of heterogeneous clouds in shortwave radiance and irradiance measurements, to be submitted to *JGR SEAC$^4$RS special issue*

**2 It is not completely clear how to use your findings for other users. How can we improve for example layer properties calculations from airborne irradiance measurements with respect to horizontal photon transport?**

The other reviewer also brought up this point. Indeed, the term "parameterization" might suggest that it can be exploited for inferring, simplifying, or correcting 3D effects, and the authors are currently working on this topic. However, the parameterization is only the first step towards this goal, and it cannot (yet) be translated into such immediate practical applications, although this is certainly the goal for the future. The purpose of the parameterization is to capture the relationship between net horizontal photon transport

and its spectral dependence using one main parameter (ε). The companion paper (Song et al., 2016) will look at the connections between 3D effects on irradiances and radiances. We will include this explanation in the revised version. For example, we conclude the abstract with the following statement: "Since three-dimensional effects depend on the spatial context of a given pixel in a non-trivial way, the spectral dimension of this problem may emerge as the starting point for future bias corrections." In section 6, we included this statement "Although our study was instigated by aircraft measurements, its findings are also relevant for satellite-based derivations of cloud radiative effects since the spectral perturbations $d\lambda$ propagate into observed radiances (Song et al., 2016). This may be exploited in future applications for deriving correction terms for 3D radiative effects via their spectral signature." We hope this clarifies the purpose of the parameterization.

Specific comments:
**1 In the last sentence of the abstract the authors mention a companion paper. It is not necessary to refer to this publication in the abstract. Rather the authors should give an example how and where the parameterization can be applied for other users.**

We made this change. We also added an outlook as final sentence in the abstract, which makes clear how the correlations and the parameterization may be used in the future ("Since three-dimensional effects depend on the spatial context of a given pixel in a non-trivial way, the spectral dimension of this problem may emerge as the starting point for future bias corrections.") At this point, the parameterization is useful to understand measurements of horizontal photon transport in inhomogeneous scenes, and can essentially be used as "fitting function" for the spectra with the free parameter ε. We will include a statement to this effect in the next paper (Song et al., 2016). In fact, this has already been done in the Ph.D. thesis (Song, 2016) which will become available for download on 8/18/2016. Once this happens, we will include a link and reference in this paper.

**2 (p3, l7) "can assume similar values as the absorbed irradiance"; Comparing the apparent absorption shown in Fig. 4a (500 nm) and 4b (1600 nm) in Schmidt et al. (2010) I identify the more the same magnitude than similar values. It's still a variable factor between the numbers. Use "same magnitude" instead "similar values". In addition, the authors should give reasons for smaller H-values in the NIR.**

We changed the wording slightly to make this distinction. We actually did not say that H values are smaller in the NIR; we only compared H (VIS) to A (NIR). A more thorough discussion is given by Schmidt et al. (2010).

**3 (p3,l20ff) The wavelength dependence of horizontal photon transport is mentioned here. Could you give a more detailed literature review on this since it is crucial for the entire manuscript?**

There have been many studies on the wavelength dependence of 3D effects in *radiance*, and the manuscript cites a small sub-set of these in Section 5 (Wen et al., 2007; Marshak et al., 2008; Varnai and Marshak, 2009), at which point the connection to the irradiances

is made. It reads as follows: "Remote sensing studies (e.g., Marshak et al., 2008; Várnai and Marshak, 2009) had previously established that the above-mentioned *radiance enhancement* for clear-sky pixels near clouds was associated with "apparent bluing," and proposed molecular scattering as the underlying cause for this spectral dependence." Following the reviewer's suggestion, we did add two additional studies pertaining to radiances (Marshak et al., 2014; Kassianov and Ovtchinnikov, 2008) further up in the text, which now reads: "For the extreme case of zero cloud optical thickness, the effect of horizontal photon transport had previously been observed as clear-sky radiance enhancement in the vicinity of clouds (Wen et al., 2007; Kassianov and Ovtchinnikov, 2008; Várnai and Marshak, 2009; Marshak et al., 2014)."

Unfortunately, studies for *irradiances* are rare, and the only ones that the authors were aware of (Ackerman and Cox, 1981; Marshak et al. 1999; Kindel et al., 2011) had been cited. However, the most relevant study (the one by Kassianov) had only been included as a footnote, and we moved it into the body of the text at the location commented on by the reviewer.

**4 (p4, l2-15) The paragraph is a mixture of outline and outlook (l6-9). Please strengthen the content. A structure of the paper is already described in the last paragraph of the introduction. Therefore the idea of the paper should be presented before (performing 3D and 1D simulations with a measured cloud data set, identifying H and it's spectral behavior, . . .) without prejudging the results.**

Thank you for catching this, we agree. We deleted the lines in question (L6-9, also 13-15). We also shortened the introduction in general.

**5 (p5, l18) Eq. (3) states the spectral absorptance. Add here directly, that these layer properties are valid for homogeneous conditions without horizontal photon transport. The reader might be confused otherwise because Eq. (3) contradicts Eq. (1) without this restriction (as noted only on p.6, l5-7).**

Thank you for this helpful comment. We made the reader aware of the difference between (1) and (3) by pre-ambling the formula with this statement: "For **homogeneous conditions ($H=0$)**, this can be quantified in terms of the layer property absorptance".

**6 (p8, l8-12) This paragraph gives an outlook. Better put this at the end of the manuscript.**

This statement (l8-12) was deleted.

**7 (p9, l4-6) As stated by the authors using height-constant effective radii has an effect on the vertical distribution of the phase functions which probably differ from reality. Why does the phase function don't affect the 3D radiative transfer? Changes of the phase function result in changes of the scattering direction. Maybe this is not as relevant as for radiance simulations. Please clarify.**

We agree that this simplification undoubtedly has an effect, and we only made this

simplification lacking better data. It is true that this would be a bigger problem for radiances than for irradiances because of the hemispherical integration. Luckily, this paper is basically a *modeling* study, albeit based on observations. We preferred actual imagery data to idealized cloud fields, which arguably could also have worked to carry out the study. Whether or not our calculations actually depicted the truth is therefore not as relevant for the message of the paper. This is different in the follow-on study (Song et al., 2016) where we used actual irradiance measurements to validate the model output.

**8 (p9, l8) Please define WC.**

Done (it's water content, not a sanitary facility ☺).

**9 (p9, l17) Please justify the choice of spatial resolution (with respect to typical spatial scales of radiative smoothing).**

The chosen resolution is certainly not fine enough to reproduce radiative smoothing in the radiance fields, but that was also not the point of the paper, which focuses on radiative energy budget quantities instead. The finest scale that is usually considered in such studies is 1km. We modified the sentence in question to "The resulting cloud field was gridded to a resolution of 0.5 km horizontally **(similar to the MODIS pixel size of some channels)** and 1.0 km vertically (chosen larger than the mismatch between CRS and MAS in cloud top height)," in order to convey our motivation for 0.5 km as spatial resolution. Undoubtedly, a finer resolution would be better, but it would have been computationally prohibitive to achieve appropriate signal-to-noise level for each of the pixels.

**10 (p11, l16) What will be generalized? The solar position?**

We modified the sentence as follows: "The scene parameters (solar geometry, surface albedo, cloud properties) will be generalized in future work (Song, 2016)."

**11 (p12, l8-11) The enhancement of radiance in the vicinity of clouds is mentioned here. Can you cite also papers dealing with the enhancement of irradiances? Add also the fact that this effect is wavelength-dependent.**

We added more references at this point (Wen et al., 2007; Kassianov and Ovtchinnikov, 2008; Várnai and Marshak, 2009; Marshak et al., 2014). See also our response for comment #3 regarding the wavelength dependence. We did not really mention the enhancement of *irradiances* in this context; this has been done in numerous other studies (including two of our own, Schmidt et al., 2007; 2009). We didn't cite these here because we wanted to keep this focused at the wavelength dependence. Note that the Kassianov paper is the only one (to our knowledge) besides the Ackerman and Cox paper which addresses this topic.

 #12 (p13, l15) Could you insert the linear fit in Fig. 3a?

Done, and we added a statement later on (following the discussion of Equation (12)) that a linear fit is less accurate than the spectrally dependent parameterization that we

developed later on: "This is more accurate than the derivation of the slope from a linear fit to the spectrum as used for Fig. 3, which, due to the non-linearity of the spectral dependence, differs from that of the tangent if finite wavelength intervals are used."

**13 (p13, l24) "pixel-to-pixel radiation exchange" → Please add "horizontal" here. There is of course a vertical exchange of photons.**

Done.

**14 (p18, l16-19) "Eq. (1) suggests..." In my opinion these two sentences do not contribute significantly to the context of this section. Referring to transmittance here somehow interrupts the flow of the discussion on spatial aggregation.**

We deleted these two sentences.

**15 (p20, l20) Please motivate the restriction of conservative scattering here, otherwise the missing absorption term might confuse the reader.**

We did remind the reader in a few places that we are only talking about wavelengths where clouds do not absorb; the general equations including A are only used to motivate our study in the introduction. However, the other reviewer also commented on the potential confusion on p20/L20, and to make it clear that we are making the simplification of A=0, we modified the text as follows: "Juxtaposing energy conservation for a horizontally homogeneous atmosphere ($T$IPA + $R$IPA = 1) with Eq. (1) for conservative scattering (**A=0, therefore** $T$3D + $R$3D = 1 – $H$) yields the plausible relationship..." We will turn our attention to wavelengths where clouds do absorb in a future study.

**16 (Sect. 8, first paragraph) To make sure that the equations are valid only for a specific wavelength range, the index "$\lambda$" would be helpful for H, R, T,...**

We preceded the formulae with this statement "For any atmospheric column, $H$ is connected to $R$ and $T$ through Eq. (1) and manifests itself in a transmittance and reflectance bias ($\lambda$ index omitted):" to indicate that the following discussion addresses a range of wavelengths (with conservative scattering, as later explained).

**17 (p23, l4, l10) If you give numbers here then you have to mention that these numbers are case specific with respect to surface albedo and solar position.**

We modified the text as follows to clarify the scene dependence of $\varepsilon$: "$\varepsilon = 0.7 \pm 0.1$ **for the scene we studied**" on p23,l10. As for the x parameter, it is not scene dependent. We did not make a strong statement about this in this manuscript because more scenes will need to be studied, but it is plausible that x~4 would not change much from scene to scene, whereas $\varepsilon$ depends on scene parameters such as surface albedo. We pointed out the need for further studies on what drives $\varepsilon$ in the following question (conclusions): "How does the discovered correlation and the constant of proportionality in its parameterization,

$\varepsilon$, depend on scene parameters such as solar zenith and azimuth angle, surface albedo (magnitude and spectral dependence), and cloud morphology and microphysics? What "drives" the parameter $\varepsilon$?" This question is addressed by Song (2016, chapter 4), and the content of this dissertation chapter will likely be published as a stand-alone paper at a later time (probably combined with the generalization to NIR wavelengths).

**18 (Sect. 9) Be more consistent with using indices for H. For example p.23, l.16: Is it H or $H_0$ or $H_\lambda$ which has to be known?**

We attempted to follow this suggestion and went through the indexing in the manuscript. In this particular place, we changed as follows: "Once $\varepsilon$ is established for a given cloud scene, the spectral perturbations associated with horizontal photon transport can be derived for each pixel if the value of $H_0$ is known. Conversely, if the spectral shape of $H_\lambda$ is known at one wavelength, its magnitude can easily be inferred for the whole spectrum."

**19 (Fig3b) Is there any reason for the increasing scatter[] of 3D-based $S_0 - H_0$ correlation for negative slopes?**

Great question; there are two parts to this: (a) the asymmetry between the (negative) minimum of $H_0$ and the (positive) maximum [probably not what the reviewer referred to] and (b) the increasing variability of $S_0$ for a fixed (negative) $H_0$.

Regarding (a): In the domain average, $\langle H_0 \rangle = 0$ despite the asymmetry. This is because fewer cloudy pixels with high values of $H_0$ balance a larger number of clear or low-optical thickness pixels with smaller (negative) values of $H_0$.

Regarding (b): We don't have a very good understanding of this yet, but the likely explanation is that for pixels that are clear or have low optical thickness, the spectral signature associated with horizontal photon transport may be affected by additional processes that are not captured by the simplified mechanism as presented in Figure 5. For example, for an optical thickness <4, the partial compensation to horizontal photon transport through molecular scattering as indicated by blue arrows may become more complicated. We did not comment on this extensively and leave this to the future. We did, however, add the following statement to section 6: "Note that below $\tau \approx 4$, directly transmitted radiation dominates the downwelling irradiance, and the cloud may not act as a "diffuser" as shown in Fig. 5. The direction of the green arrows is then along the direct beam." This effect is most likely the cause for the deviation from the correlation that the reviewer observed.

Technical comments:

1) Please remove footnotes

Done.

2) Check that symbols in figures have italic format.

Done, figures were replaced.

3) (p12, l25) Figs. → Fig. 4): (p13, l25) "H" → "$H_0$" 5): (p14, l3) "$H_\lambda$" → "$H_\lambda$" (italic)

All done, thanks.